# BEYOND THE RAINBOW: HIGH PERFORMANCE DEEP REINFORCEMENT LEARNING ON A DESKTOP PC

## ABSTRACT

Rainbow Deep Q-Network (DQN) demonstrated combining multiple independent enhancements could significantly boost a reinforcement learning (RL) agent's performance. In this paper, we present "Beyond The Rainbow" (BTR), a novel algorithm that integrates six improvements from across the RL literature to Rainbow DQN, establishing a new state-of-the-art for RL using a desktop PC, with a human-normalized interquartile mean (IQM) of 7.6 on Atari-60. Beyond Atari, we demonstrate BTR's capability to handle complex 3D games, successfully training agents to play Super Mario Galaxy, Mario Kart, and Mortal Kombat with minimal algorithmic changes. Designing BTR with computational efficiency in mind, agents can be trained using a high-end desktop PC on 200 million Atari frames within 12 hours. Additionally, we conduct detailed ablation studies of each component, analyzing the performance and impact using numerous measures.

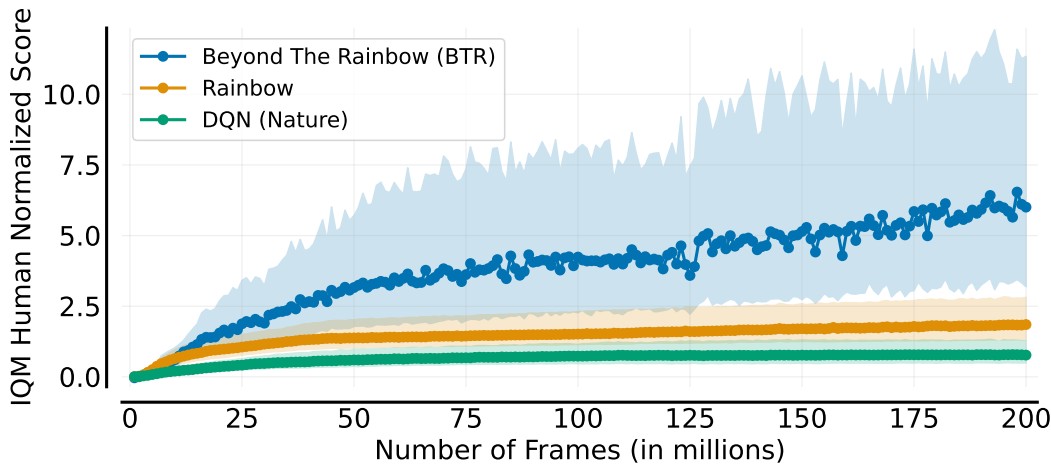

Figure 1: Interquartile mean human-normalized performance for BTR against other RL algorithms on the Atari-60 benchmark. The results for DQN and Rainbow DQN are those reported in RLiable (Agarwal et al., 2021). Shaded areas show 95% confidence intervals bootstrapped over tasks, with BTR using 1 seed. For box plots and performance profiles, see Appendix B.

## 1 INTRODUCTION

Deep Reinforcement Learning (RL) has achieved numerous successes in complex sequential decision-making tasks, most rapidly since Mnih et al. (2015) proposed Deep Q-Learning (DQN). With this success, RL has become increasingly popular among smaller research labs, the hobbyist community, and even the general public. However, recent state-of-the-art approaches (Schrittwieser et al., 2020; Badia et al., 2020a; Hessel et al., 2021; Kapturowski et al., 2022) are increasingly out of reach for those with more limited compute resources, either in terms of the required hardware or the walltime necessary to train a single agent. This is a unique issue in RL compared to natural language

processing or image recognition which have foundation models that can be efficiently fine-tuned for a new task or problem (Lv et al., 2023). Meanwhile, RL agents must be trained afresh for each environment. Therefore, the development of powerful RL algorithms that can be trained quickly on inexpensive hardware is crucial for smaller research labs and the hobbyist community.

These concerns are not new. Ceron & Castro (2021) highlighted that Rainbow DQN (Hessel et al., 2018) required 34,200 GPU hours (equivalent to 1435 days) of training, making the research impossible for anyone except a few research labs, with more recent algorithms exacerbating this problem. Recurrent network architectures (Horgan et al., 2018), high update to sample ratio (D'Oro et al., 2022), and the use of world-models and search-based techniques (Schrittwieser et al., 2020) all increase the computational resources necessary to train agents, many using distributed approaches requiring multiple CPUs and GPUs (or TPUs), or requiring numerous days and weeks to train a single agent. These features have dramatically decreased RL's accessibility.

For this purpose, we develop "Beyond the Rainbow" (BTR), taking the same principle as Rainbow DQN Hessel et al. (2018), selecting 6 previously independently evaluated improvements and combining them into a singular algorithm (Section 3). These components were chosen for their performance qualities or to reduce the computational requirements for training an agent. As a result, BTR sets a new state-of-the-art score for Atari-60 (Bellemare et al., 2013) (excluding recurrent approaches) with an Interquartile Mean (IQM) of 7.6[1] using a single desktop machine in less than 12 hours, and outperforms Rainbow DQN on Procgen (Cobbe et al., 2020) in less than a fifth of the walltime (Section 4.1). Further, we demonstrate BTR's potential by training agents to solve three modern 3D games for the first time, Mario Kart Wii, Super Mario Galaxy and Mortal Combat, that each contain complex mechanics and graphics (Section 4.2). To verify the effectiveness and effect of the six improvements to BTR, in Section 5.1, we conduct a thorough ablation of each component, plotting their impact on the Atari-5 environments and in Section 5.2, we utilise seven different measures to analyse the component's impact on the agent's policy and network weights. This allows us to more precisely understand how the components impact BTR beyond performance or walltime.

> In summary, we make the following contributions to state-of-the-art RL.
>
> - **High Performance (Section 4.1) -** BTR outperforms the state-of-the-art for non-recurrent RL on the Atari-60 benchmark, with an IQM of 7.6 (compared to Rainbow DQN's 1.9), outperforming humans on 52/60 games. Furthermore, BTR outperforms Rainbow DQN with Impala on the Procgen benchmark despite using a smaller model and 80% less walltime.
>
> - **Modern Environments (Section 4.2) -** Testing beyond Atari, we demonstrate BTR can train agents for 3 modern games: Super Mario Galaxy (final stage), Mario Kart Wii (Rainbow Road), and Mortal Combat (Endurance mode). These environments contain 3D graphics and complex physics and have never been solved using RL.
>
> - **Computationally Accessible (Figure 5) -** Using a high-end desktop PC, BTR trains Atari agents for 200 million frames in under 12 hours, significantly faster than Rainbow DQN's 35 hours. This increases RL research's accessibility for smaller research labs and hobbists without the need for GPU clusters or excessive walltime.
>
> - **Component Impact Analysis (Section 5) -** We conduct thorough ablations investigating BTR without each component in terms of performance and other measures. We discover that BTR widens action gaps (reducing the effects of approximation errors), is robust to observation noise, and reduces neuron dormancy and weight matrix norm (shown to improve plasticity throughout training).

## 2 BACKGROUND

Before describing BTR's extensions, we outline standard RL mathematics, how DQN is implemented, and Rainbow DQN's extensions.

---

[1] All reported IQM scores use the best single evaluation for each environment throughout training as is standard, rather than the agent's score at 200 million, hence the discrepancy between the overall score and Figure 1.

## 2.1 RL Problem Formulation

We adopt the standard formulation of RL (Sutton & Barto, 2018), described as a Markov Decision Process (MDP) defined by the tuple $(\mathcal{S}, \mathcal{A}, \mathcal{P}, \mathcal{R})$, where $\mathcal{S}$ is the set of states, $\mathcal{A}$ is the set of actions, $\mathcal{P} : \mathcal{S} \times \mathcal{A} \to \Delta(\mathcal{S})$ is the stochastic transition function, and $\mathcal{R} : \mathcal{S} \times \mathcal{A} \to \mathbb{R}$ is the reward function. The agent's objective is to learn a policy $\pi : S \to \Delta(\mathcal{A})$ that maximizes the expected sum of discounted rewards $\mathbb{E}_\pi[\sum_{t=0}^{\infty} \gamma^t r(s_t, a_t)]$, where $\gamma \in [0, 1)$ is the discount rate.

## 2.2 Deep Q-Learning (DQN)

One popular method for solving MDPs is Q-Learning (Watkins & Dayan, 1992) where an agent learns to predict the expected sum of discounted future rewards for a given state-action pair. To allow agents to generalize over states and thus be applied to problems with larger state spaces, Mnih et al. (2013) successfully combined Q-Learning with neural networks. To do this, training minimizes the error between the predictions from a parameterized network $Q_\theta$ and a target defined by

$$r_t + \gamma \max_{a \in A} Q_{\theta'}(s_{t+1}, a) \, , \tag{1}$$

where $Q_{\theta'}$ is an earlier version of the network referred to as the target network, which is periodically updated from the online network $Q_\theta$. The data used to perform updates is gathered by sampling from an Experience Replay Buffer (Lin, 1992), which stores states, actions, rewards, and next states experienced by the agent while interacting with the environment.

## 2.3 Rainbow DQN and Improvements to DQN

In collecting 6 different improvements to DQN, Rainbow DQN (Hessel et al., 2018) proved cumulatively that these improvements could achieve a greater performance than any individually. We briefly explain the individual improvements, ordered by performance impact, most of which are preserved within BTR (see Table 1), for more detail, we refer readers to the extension's respective papers:

1. **Prioritized Experience Replay -** To select training examples, DQN sampled uniformly from an Experience Replay Buffer, assuming that all examples are equally important to train with. Schaul et al. (2015) proposed sampling training examples proportionally to their last seen absolute temporal difference error, encouraging more training on samples for which the network most inaccurately predicts their future rewards.

2. **N-Step -** Q-learning utilizes bootstrapping to minimize the difference between the predicted value and the resultant reward plus the maximum value of the next state (Eq. 1). N-step (Sutton et al., 1998) reduces the reliance on this bootstrapped next value by considering the next $n$ rewards and observation in $n$ timesteps (Rainbow DQN used $n = 3$).

3. **Distributional RL -** Due to the stochastic nature of RL environments and agent's policies, Bellemare et al. (2017) proposed learning the return distribution rather than scalar expectation; this was done through modelling the return distributions using probability masses and the Kullbeck-Leibler divergence loss function.

4. **Noisy Networks -** Agents can often insufficiently explore their environment resulting in sub-optimal policies. Fortunato et al. (2017) added parametric noise to the network weights, causing the model's outputs to be randomly perturbed, increasing exploration during training, particularly for states where the agent has less confidence.

5. **Dueling DQN -** The agent's Q-value can be rewritten as the sum of state-value and advantage ($Q(s, a) = V(s) + A(s, a)$). Looking to improve action generalisation, Wang et al. (2016) split the hidden layers into two separate streams for the value and advantage, recombining them with $Q(s, a) = V(s) + (A(s, a) - \frac{1}{|\mathcal{A}|} \sum_{a'} A(s, a'))$.

6. **Double DQN -** In selecting the next observation's maximum Q-value (Eq. 1), this can frequently overestimate the target's Q-value, negatively affecting the agent's performance. To reduce this overestimation, Van Hasselt et al. (2016) propose utilising the online network rather than the target network to select the next action when forming targets, defined as:

$$r_t + \gamma Q_{\theta'}(s_{t+1}, \arg\max_{a \in A} Q_\theta(s_{t+1}, a)) \, . \tag{2}$$

## 3 BEYOND THE RAINBOW - EXTENSIONS AND IMPROVEMENTS

Building on Rainbow DQN (Hessel et al., 2018), BTR includes 6 more improvements undiscovered in 2018.[2] Additionally, as hyperparameters are critical to agent performance, Section 3.2 discusses key hyperparameters and our choices. In the appendices, we include a table of hyperparameters, a figure of the network architecture and the agent's loss function (Appendices C.2, D and D.2). Finally, the source code using Gymnasium (Towers et al., 2024) is included within the supplementary material to help future work build upon or utilise BTR.

Table 1: A comparison of components between Rainbow DQN (Hessel et al., 2018) and BTR.

| Added To Rainbow DQN | Same As Rainbow DQN | Removed From Rainbow DQN |
|---|---|---|
| Impala (Scale=2) | N-Step TD Learning | Double (N/A with Munchausen) |
| Adaptive Maxpooling (6x6) | Prioritized Experience Replay | C51 (Upgraded to IQN) |
| Spectral Normalisation | Dueling | |
| Implicit Quantile Networks | Noisy Networks | |
| Munchausen | | |
| Vectorized Environments | | |

### 3.1 EXTENSIONS

**Impala Architecture + Adaptive Maxpooling** - Espeholt et al. (2018) proposed a convolutional residual neural network architecture based on He et al. (2016) featuring three residual blocks[3], substantially increasing performance over DQN's three-layer convolutional network. Following Cobbe et al. (2020), we scale the width of the convolutional layers by 2 to enhance its capabilities. We include an additional 6x6 adaptive max pooling layer after the convolutional layers (Schmidt & Schmied, 2021) found to speed up learning and support different input resolutions. Our adaptive maxpooling is identical to that of a standard 2D maxpooling layer, but can be used with any input resolution as it automatically adjusts the stride and kernel size to fit the specified output size (6x6).

**Spectral Normalisation (SN)** - To help stabilize the training of discriminators in Generative Adversarial Networks (GANs), Miyato et al. (2018) proposed Spectral Normalisation to help control the Lipschitz constant of convolutional layers. SN works to normalize the weight matrices of each layer in the network by their largest singular value, ensuring that the transformation applied by the weights does not distort the input data excessively, which can lead to instability during training. Bjorck et al. (2021) and Gogianu et al. (2021) found that SN could improve performance in RL, especially for larger networks and Schmidt & Schmied (2021) found SN reduced the number of updates required before initial progress is made.

**Implicit Quantile Networks (IQN)** - Dabney et al. (2018) improved upon Bellemare et al. (2017), used in Rainbow DQN, learning the return distribution over the probability space rather than probability distribution over return values. This removes the limit on the range of Q-values that can be expressed, and enables learning the expected return at every probability.

**Munchausen RL** - Boostrapping is a core aspect of RL; used to calculate target values (Eq. 1) with most algorithms using the reward, $r_t$, and the optimal Q-value of the next state, $Q^*$. However, since in practice the optimal policy is not known, the current policy $\pi$ is used. Munchausen RL (Vieillard et al., 2020) looks to leverage an additional estimate in the bootstrapping process by adding the scaled-log policy to the loss function (Eq. 3 where $\alpha \in [0, 1]$ is a scaling factor, $\sigma$ is the softmax function, and $\tau$ is the softmax temperature). This assumes a stochastic policy, therefore DQN is converted to Soft-DQN with with $\pi_{\theta'} = \sigma(\frac{Q_{\theta'}}{\tau})$. As Munchausen does not use argmax over the next state, Double DQN is obsolete. Munchausen RL's update rule is

$$Q_\theta(s_t, a_t) = r_t + \alpha \tau \ln \pi_{\theta'}(a_t|s_t) + \gamma \sum_{a' \in A} \pi_{\theta'}(a'|s_{t+1})(Q_{\theta'}(s_{t+1}, a') - \tau \ln(\pi_{\theta'}(a'|s_{t+1}))) . \quad (3)$$

---

[2]After the completion of our work, we additionally found Layer Normalization applied after the stem of each residual block and between dense layers to be beneficial (see Appendix I for a discussion)

[3]The network architecture is referred to as Impala due to the accompanying training algorithm IMPALA proposed in Espeholt et al. (2018)

**Vectorization** - RL agents typically take multiple steps in a single environment, followed by a gradient update with a small batch size (Rainbow DQN took 4 environment steps, followed by a batch of 32). However, taking multiple steps in parallel and performing updates on larger batches can significantly reduce walltime. We follow Schmidt & Schmied (2021), taking 1 step in 64 parallel environments with one gradient update with batch size 256 (Schmidt & Schmied (2021) took two gradient updates rather than the one we take). This results in a replay ratio (ratio of gradient updates to environment steps) of $\frac{1}{64}$. Higher replay ratios have been shown to improve performance (D'Oro et al., 2022), however we opt to keep this value low to reduce walltime.

### 3.2 HYPERPARAMETERS

Hyperparameters have repeatedly shown to have a very large impact on performance in RL (Ceron et al., 2024), thus we perform a small amount of tuning to improve performance. Firstly, how frequently the target network is updated is closely intertwined with batch size and replay ratio. We found that updating the target network every 500 gradient steps[4] performed best. Given our high batch size, we additionally performed minor hyperparameter tests using different learning rates finding that a slightly higher learning rate of $1 \times 10^{-4}$ performed best, compared to $6.25 \times 10^{-5}$ in Rainbow DQN. In Appendix C.2, we clarify the meaning of the terms frames, steps and transitions.

For many years, RL algorithms have used a discount rate of 0.99, however, when reaching high performance, lower discount rates alter the optimal policy, causing even optimally performing agents to not collect the maximum cumulative rewards. To prevent this, we follow MuZero Reanalyse (Schrittwieser et al., 2021) using $\gamma = 0.997$. For our Prioritized Experience Replay, we use the lower value of $\alpha = 0.2$, the parameter used to determine sample priority, recommended by Toromanoff et al. (2019) when using IQN. Lastly, many previous experiments used only noisy networks or $\epsilon$-greedy exploration, however, we opt to use both until 100M frames, then set $\epsilon$ to zero, effectively disabling it. We elaborate on this decision in Appendix G.

## 4 EVALUATION

To assess BTR, we test it on two standard RL benchmarks, Atari (Bellemare et al., 2013) and Procgen (Cobbe et al., 2020) in Section 4.1. Secondly, we train BTR agents for three modern games (Super Mario Galaxy, Mario Kart Wii, and Mortal Combat) with complex 3D graphics and physics in Section 4.2, never shown to be trainable with RL previously.

### 4.1 ATARI AND PROCGEN PERFORMANCE

We evaluate BTR on the Atari-60 benchmark following (Machado et al., 2018) and without life information (see Appendix J for the impact), evaluating every million frames on 100 episodes. Figure 1 plots BTR against Rainbow DQN and DQN, achieving an IQM of 7.6 compared to Rainbow DQN's 2.7 and DQN's 0.9. In comparison to human expert performance, BTR equals or exceeds them in 52 of 60. Importantly, we find that BTR appears to continue increasing performance beyond 200 million frames, indicating that higher performance is still possible with more time and data. Results tables and graphs can be found in Appendices A and B, respectively.

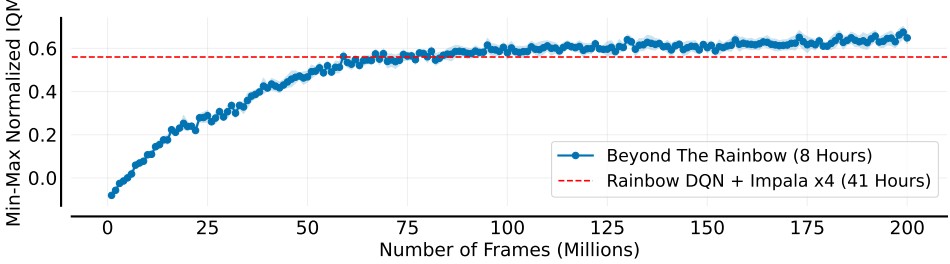

Figure 2: BTR compared to Rainbow DQN + Impala (width x4) (Cobbe et al., 2020) after 200M frames on the Procgen benchmark. Shaded areas show 95% CIs, with results averaged over 2 seeds.

---

[4]This equates to 32,000 environment steps (128,000 frames), compared to Rainbow DQN's 8,000 steps.

To further confirm BTR's performance, we benchmark on Procgen (Cobbe et al., 2020), a procedurally generated set of environments aiming to prevent overfitting to specific tasks, a prevalent problem in RL (Justesen et al., 2018; Juliani et al., 2019). The results are shown in Figure 2 with individual games in Appendix B. BTR is able to exceed Rainbow DQN + Impala's performance, despite using significantly fewer convolutional filters (which Cobbe et al. (2020) found to significantly improve performance) and using 8 hours of walltime compared to 41. These results demonstrate BTR's general learning capability across a wide range of standard RL benchmarks.

## 4.2 APPLYING BTR TO MODERN GAMES

To demonstrate BTR's capabilities beyond standard RL benchmarks, we utilised Dolphin (Dolphin-Emulator, 2024), a Nintendo Wii emulator, to train agents for a range of modern 3D games: Super Mario Galaxy, Mario Kart Wii and Mortal Combat. Using a desktop PC, we were able to train the agent to complete some of the most difficult tasks within each game. Namely, the final level in Super Mario Galaxy, Rainbow Road (a notoriously difficult track in Mario Kart Wii) and defeating all opponents in Mortal Kombat Endurance mode (for details about the environments and setup, see Appendix K). For this, BTR required minimal adjustments: first, to input image resolution, 140x114 (from Atari's 84x84) due to the game's higher resolution and aspect ratio, and second, to reduce the number of vectorized environments to 4 as a result of the games' memory and CPU requirements.

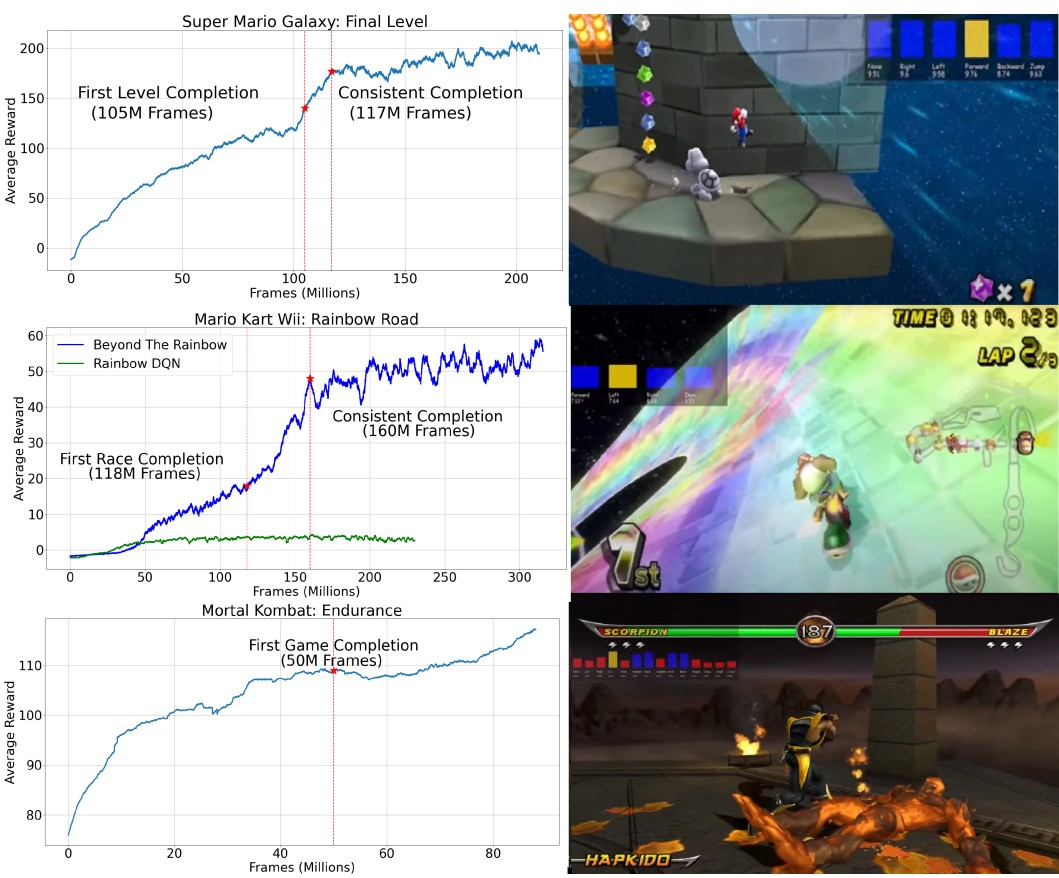

Figure 3: BTR being used to play Super Mario Galaxy (final level), Mario Kart Wii (Rainbow Road) and Mortal Kombat: Armageddon (Endurance Mode) respectively.

For all the games, BTR was able to solve the game level, including consistently finishing in first place in Mario Kart. We provide videos of our agent playing all three Wii games, in addition to the games in the Atari-5 benchmark [5].

---

[5] https://www.youtube.com/playlist?list=PL4geUsKi0NN-sjbuZP_fU28AmAPQunLoI

## 5 ANALYSIS

Given BTR's performance demonstrated in Section 4, in this Section, we ablate each component to evaluate their performance impact (Section 5.1). Using the ablated agents, we then measure numerous attributes during and after training to assess each component's impact (Section 5.2).

### 5.1 ABLATIONS STUDIES

BTR amalgamates independently evaluated components into a single algorithm. To understand and verify each's contribution, Figure 4 plot BTR's performance without each component on the Atari-5 benchmark.[6]

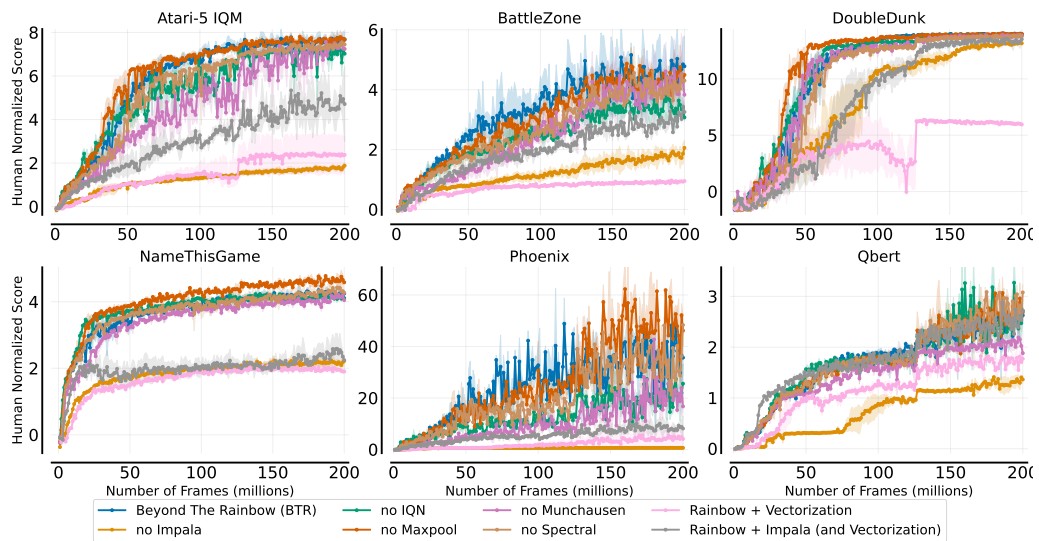

Figure 4: Individual game performance of BTR on Atari-5 with individual components removed averaged over 3 seeds. Shaded areas show 95% confidence intervals. Note the Atari-5 IQM does not use the regression procedure from Aitchison et al. (2023) due to adverse results (see Appendix L).

We find that Impala had the largest effect on performance, with the other components generally causing a less significant effect on final performance. However, when BTR's performance is compared before 200 million frames, we find Munchausen and Spectral Normalisation provide significant performance improvements (+24% and +25% at 40M frames, and +13% and +35% at 120M frames). We compare the performance of components at different stages of training in Appendix E.

For vectorization and maxpooling, while their inclusion reduces performance, we find their secondary effects crucial to keep BTR computationally accessible. Omitting vectorization increases walltime by 328% (Figure 5) by processing environment steps in parallel and taking fewer gradient steps (781,000 compared to Rainbow DQN's 12.5 million).[7] We find maxpooling makes the agent more robust to noise as discussed in Section 5.2, and decreases the model's parameters by 77%.

### 5.2 WHAT ARE THE EFFECTS OF BTR'S COMPONENTS?

To help interpret the results in Section 5.1, we measure seven different attributes of the agent either during or after training: action gaps and action swaps, linked to causing approximation errors (Bellemare et al., 2016); policy churn, which can cause excessive off-policyness (Schaul et al., 2022) and score with additional noise indicating robustness. For analyses of model weights, see Appendix F.

---

[6] Due to computational resources required to evaluate each component on 60 environments, Aitchison et al. (2023) proposes a subset of 5 games that closely correlate with the performance across all 60.

[7] Another consequence of removing vectorization is using smaller batches, which Obando Ceron et al. (2024) finds improves exploration, possibly explaining our results found in Qbert.

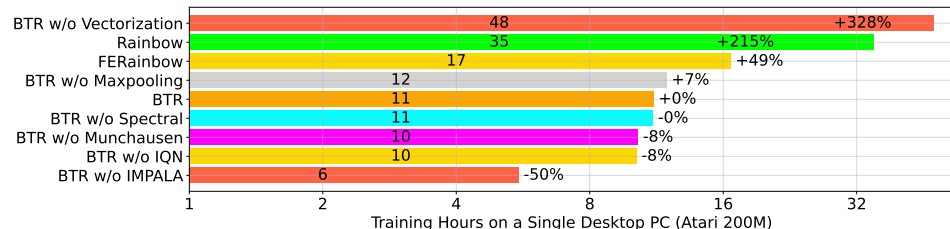

Figure 5: Walltime of BTR on a desktop PC with components removed, compared with Hessel et al. (2018) and Schmidt & Schmied (2021). For hardware details, see Appendix H.

For why Impala contributes to performance so strongly, we find that without BTR's other components, Impala exhibits a notable drawback, learning a highly noisy and unstable policy. Table 2, demonstrates that without IQN and Munchausen the agent experiences very low action gaps (absolute Q-value difference between the highest two valued actions), causing the agent to swap its argmax action almost every other step. This is likely to result in approximation errors altering the policy and causing a high degree of off-policyness in the replay buffer. This is particularly detrimental in games requiring fine-grained control, such as *Phoenix* where the agent needs to narrowly dodge many projectiles, reflected in BTR's performance without these components.

Furthermore, we find that maxpooling is useful in dealing with noisy environments. To test this, we evaluate the performance of BTR's ablations when taking different quantities of $\epsilon$-actions, and find these two components prevent performance from dropping substantially 2. Lastly, we find Munchausen and IQN to have a significant impact on Policy Churn (Schaul et al., 2022), with Munchausen reducing it by 6.4% and IQN increasing it by 3.3%. As a result, when these components are used together, they appear to reach a level of policy churn which does not harm learning and potentially provides some exploratory benefits.

Table 2: Comparison of policy churn, action gaps, actions swaps and evaluation performance with different quantities of $\epsilon$-actions and color jitter (both only applied for evaluation). All measurements use the final agent, trained on 200 million frames, for Atari *Phoenix*, averaged over 3 seeds. Action Gap is the average absolute Q-value difference between the highest two valued actions. % Actions Swap is the percentage of times the agent's argmax action has changed from the last timestep. Policy churn is the percentage of states which the agent's argmax action has changed on after a single gradient step. Color jitter applies a random 10% change to the brightness, saturation and hue of each frame. For associated error with these values, please see Appendix F.

| Category | BTR | w/o Munchausen | w/o IQN | w/o SN | w/o Impala | w/o Maxpool |
|---|---|---|---|---|---|---|
| Action Gap | 0.281 | 0.056 | 0.175 | 0.298 | 0.313 | 0.280 |
| % Action Swaps | 33.4% | 45.8% | 42.2% | 39.7% | 28.6% | 39.2% |
| Policy Churn | 3.8% | 10.2% | 0.5% | 2.9% | 4.2% | 3.9% |
| Score ColorJitter | **206k** | 80k | 93k | 178k | 5k | 172k |
| Score $\epsilon = 0.03$ | **98k** | 47k | 57k | 79k | 5k | 94k |
| Score $\epsilon = 0.01$ | **208k** | 79k | 110k | 181k | 5k | 167k |
| Score $\epsilon = 0$ | 397k | 279k | 199k | 356k | 5k | **489k** |

# 6    RELATED WORK

The most similar work to BTR, developing a computationally-limited non-distributed RL algorithm, is "Fast and Efficient Rainbow" (Schmidt & Schmied, 2021). They optimised Rainbow DQN to maximise performance for 10 million frames through parallelizing the environments and dropping C51 along with hyperparameter optimisations. This differs from our goals of producing an algorithm that scales across training regimes (up to 200 million frames) and domains (Atari, Procgen, Super Mario Galaxy, Mario Kart and Mortal Combat), resulting in different design decisions.

For less computation-limited approaches, Ape-X (Horgan et al., 2018) was the first to explore highly distributed training, allowing agents to be trained on a billion frames in 120 hours through using 100 CPUs. Following this, Kapturowski et al. (2018) proposed R2D2 using a recurrent neural network, increasing sample efficiency but slowing down gradient updates by 38%. Agent57 (Badia et al., 2020a) was the first RL agent to achieve superhuman performance across 57 Atari games, though required 90 billion frames. MEME (Kapturowski et al., 2022), Agent57's successor, focused on achieving superhuman performance within the standard 200 million frames limit, achieved by used a significantly higher replay ratio and larger network architecture. Most recently, Dreamer-v3 (Hafner et al., 2023) used a 200 million parameter model requiring over a week of training, achieving similar results as MEME. We detail some of the key differences between BTR, MEME and Dreamer-v3 in Table 3. While these approaches perform equally to or better than BTR, all are inaccessible to smaller research labs or hobbyists due to their required computational resources and walltime. Therefore, while these algorithms have important research value demonstrating the possible performance of RL agents, performative algorithms with a lower cost of entry, like BTR, are a necessary component for RL to become widely applicable and accessible.

Table 3: Comparison of performance, walltime, observations and complexity of different algorithms.

| Category | BTR | MEME | Dreamer-v3 |
|---|---|---|---|
| A100 GPU Days | 0.9 | Not Reported | 7.7 |
| Recurrent? | No (4 stacked frames) | Yes | Yes |
| Learns from? | Single Transitions | Trajectories (length 160) | Trajectories (length 64) |
| World Model | No | No | Yes |
| Parameters | 2.9M | Not Reported ($\approx > 20$M) | 200M |
| Observation Shape | 84x84 | 210x160 | 64x64 |
| Gradient Steps | 781K | 3.75M | 1.5M |
| Atari-60 IQM | 7.6 | 9.6 | 9.6 |

## 7    CONCLUSION AND FUTURE WORK

We have demonstrated that, once again, independent improvements from across Deep Reinforcement Learning can be combined into a single algorithm capable of pushing the state-of-the-art far beyond what any single improvement is capable of. Importantly, we find that this can be accomplished on desktop PCs, increasing the accessibility of RL for smaller research labs and hobbyists.

We acknowledge that there are many more promising improvements we were not able to include in BTR, leaving room for more future work in a few years to create even stronger integrated agents. For example, BTR does not add an explicitly exploration component, resulting in it struggling in hard-exploration tasks such as *Montezuma's Revenge*; therefore, mechanisms used in Never Give Up (Badia et al., 2020b) or other components may prove useful. Section 5.1 found that the neural network's core architecture, Impala, had the largest impact on performance, an area we believe is generally underappreciated in RL. Previous work (Kapturowski et al., 2018) has incorporated recurrent models enhancing performance, though we are uncertain how this can be incorporated into BTR without affecting its computational accessibility.

## 8 ETHICS AND REPRODUCIBILITY STATEMENTS

Our work does not involve human subjects or methodologies with direct ethical concerns such as discrimination, bias, or privacy violations. Additionally, we have no conflicts of interest, sponsorship issues, or violations of legal or research integrity were present during the development of this research. However, we acknowledge that by improving the accessibility and performance of Reinforcement Learning (RL), our contributions may inadvertently provide more powerful tools to malicious actors, thus, we urge the research community to remain vigilant regarding these issues.

To ensure reproducibility, we provide a detailed background of the work we build upon and clearly explain all changes made to the base algorithm. Furthermore, Appendix C.2 provides all relevant environmental and algorithmic hyperparameters needed to reproduce our work. Additionally, we provide clarity about often misunderstood terms (Appendix C.3), a detailed architecture diagram (Appendix D) and the exact hardware we tested our algorithms on (Appendix H). Most importantly, we provide BTR's code within the supplementary material. Lastly, we provide many details regarding the Wii games tested BTR on, including the minor changes from BTR, how the environment was setup and the reward functions used.

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

# A  FULL RESULTS TABLES

Table A1: Maximum scores obtained during training (averaged over 100 episodes and all performed using random seeds) after 200M Frames on the Atari-60 benchmark. Fast & Efficient Rainbow DQN and Munchausen-IQN refer to Schmidt & Schmied (2021) and (Vieillard et al., 2020) respectively. FE-Rainbow uses Life Information (See Appendix J), only 10M frames, and has missing games so metrics are based on existing games.

| Game | Random | Human | DQN (Nature) | Rainbow | M-IQN | FE-Rainbow | BTR |
|---|---|---|---|---|---|---|---|
| AirRaid | 400 | 1000 | 7523 | 12472 | 19111 | | **53543** |
| Alien | 227 | 7127 | 2354 | 3610 | 4249 | 12508 | **19149** |
| Amidar | 5 | 1719 | 1268 | 2390 | 1653 | 2071 | **17807** |
| Assault | 222 | 742 | 1526 | 3490 | 6014 | 10709 | **19384** |
| Asterix | 210 | 8503 | 2803 | 16547 | 42615 | 346758 | **593650** |
| Asteroids | 719 | 47388 | 846 | 1494 | 1666 | 12345 | **169272** |
| Atlantis | 12850 | 29028 | 843372 | 791393 | 866810 | 812825 | **899104** |
| BankHeist | 14 | 753 | 560 | 1070 | 1305 | 1411 | **1598** |
| BattleZone | 2360 | 37187 | 18425 | 40316 | 50501 | 112652 | **168340** |
| BeamRider | 363 | 16926 | 5203 | 6084 | 12322 | 26398 | **138102** |
| Berzerk | 123 | 2630 | 467 | 832 | 719 | 3388 | **6703** |
| Bowling | 23 | 160 | 30 | 43 | 23 | 40 | **47** |
| Boxing | 0 | 12 | 79 | 98 | 99 | 99 | **100** |
| Breakout | 1 | 30 | 92 | 109 | 241 | 537 | **676** |
| Carnival | 380 | 4000 | 5111 | 4523 | 5588 | | **6031** |
| Centipede | 2090 | 12017 | 2378 | 6595 | 4425 | 8368 | **76242** |
| ChopperCommand | 811 | 7387 | 2722 | 13029 | 551 | 4208 | **980233** |
| CrazyClimber | 10780 | 35829 | 103549 | 146262 | **146419** | 140712 | 140723 |
| DemonAttack | 152 | 1971 | 5437 | 17411 | 63143 | 131657 | **135447** |
| DoubleDunk | -18 | -16 | -5 | 22 | 21 | -1 | **23** |
| ElevatorAction | 0 | 3000 | 408 | 79372 | **89237** | | 82669 |
| Enduro | 0 | 860 | 642 | 2165 | 2247 | 2266 | **2352** |
| FishingDerby | -91 | -38 | -1 | 42 | 54 | 42 | **56** |
| Freeway | 0 | 29 | 26 | 33 | 33 | **34** | 33 |
| Frostbite | 65 | 4334 | 482 | 8309 | 9419 | 5282 | **19331** |
| Gopher | 257 | 2412 | 5440 | 9987 | 23310 | 25606 | **99739** |
| Gravitar | 173 | 3351 | 209 | 1249 | 1105 | 2107 | **5284** |
| Hero | 1027 | 30826 | 15766 | 46290 | **25555** | 15377 | 21559 |
| IceHockey | -11 | 0 | -6 | 0 | 11 | 6 | **38** |
| Jamesbond | 29 | 302 | 671 | 995 | 1526 | | **29828** |
| JourneyEscape | -18000 | -1000 | -3300 | -1096 | -806 | | **5166** |
| Kangaroo | 52 | 3035 | 10744 | 13005 | 10704 | 11498 | **13849** |
| Krull | 1598 | 2665 | 6029 | 4368 | 10309 | 10324 | **11123** |
| KungFuMaster | 258 | 22736 | 22397 | 27066 | 25588 | 27444 | **54330** |
| MontezumaRevenge | 0 | 4753 | 0 | **500** | 0 | 0 | 0 |
| MsPacman | 307 | 6951 | 3431 | 3989 | 5630 | 5981 | **11493** |
| NameThisGame | 2292 | 8049 | 7549 | 8900 | 12440 | 19819 | **28360** |
| Phoenix | 761 | 7242 | 4993 | 8800 | 5315 | 60954 | **350722** |
| Pitfall | -229 | 6463 | -45 | -27 | -32 | -1 | **0** |
| Pong | -20 | 14 | 16 | 20 | 19 | **21** | 20 |
| Pooyan | 500 | 1000 | 3452 | 4344 | 13096 | | **24279** |
| PrivateEye | 24 | 69571 | 1113 | **21353** | 100 | 253 | 100 |
| Qbert | 163 | 13455 | 9801 | 18332 | 13159 | 25712 | **39484** |
| Riverraid | 1338 | 17118 | 9725 | 20675 | 16143 | | **24585** |
| RoadRunner | 11 | 7845 | 38430 | 55104 | 60370 | 81831 | **590236** |
| Robotank | 2 | 11 | 59 | 67 | 71 | 70 | **83** |
| Seaquest | 68 | 42054 | 2416 | 9590 | 23885 | 63724 | **409991** |
| Skiing | -17098 | -4336 | -16281 | -29268 | -10404 | -22076 | **-9131** |
| Solaris | 1236 | 12326 | 1478 | 1686 | 1835 | 2877 | **8198** |
| SpaceInvaders | 148 | 1668 | 1797 | 4455 | 10810 | 28098 | **53863** |
| StarGunner | 664 | 10250 | 48498 | 57255 | 64875 | 310403 | **574106** |
| Tennis | -23 | -8 | -3 | 0 | 0 | 15 | **23** |
| TimePilot | 3568 | 5229 | 3704 | 11959 | 14600 | 31333 | **110981** |
| Tutankham | 11 | 167 | 103 | 244 | 205 | 167 | **314** |
| UpNDown | 533 | 11693 | 8797 | 37936 | 197043 | | **397875** |
| Venture | 0 | 1187 | 13 | **1537** | 978 | 437 | 0 |
| VideoPinball | 0 | 17667 | 38720 | 460245 | 508012 | 269619 | **589065** |
| WizardOfWor | 563 | 4756 | 1473 | 7952 | 11352 | 15518 | **50828** |
| YarsRevenge | 3092 | 54576 | 23963 | 46456 | 106929 | 98908 | **177430** |
| Zaxxon | 32 | 9173 | 4471 | 14983 | 14286 | 18832 | **47096** |
| IQM (↑) | 0.000 | 1.000 | 0.771 | 1.852 | 2.181 | ≈ 2.769 | **7.572** |
| Median (↑) | 0.000 | 1.000 | 0.731 | 1.506 | 1.559 | ≈ 1.906 | **4.695** |
| Mean (↑) | 0.000 | 1.000 | 2.261 | 4.152 | 5.260 | ≈ 7.700 | **19.775** |
| Optimality Gap (↓) | 0.000 | 1.000 | 0.407 | 0.200 | 0.224 | ≈ 0.180 | **0.097** |
| Best | - | - | 0 | 3 | 3 | 2 | **52** |
| >Human | - | - | 22 | 43 | 34 | 38 | **52** |
| Surround | 7 | -10 | | | | | **10** |
| Defender | 2875 | 18689 | | | | 169929 | **461380** |

Table A2: Maximum scores obtained during training (averaged over 100 episodes and all performed 3 random seeds) after 200M Frames on the Atari-5 Environment, compared against other non-recurrent non-distributed algorithms. FE-Rainbow refers to Fast and Efficient Rainbow DQN (Schmidt & Schmied, 2021), and M-IQN refers to Munchausen-IQN (Vieillard et al., 2020). Metrics do not use the recommended regression procedure, as explained in Appendix L.

| Game | Random | Human | Rainbow DQN (Dopamine) | Rainbow DQN (Full) | M-IQN | FE-Rainbow | BTR |
|---|---|---|---|---|---|---|---|
| BattleZone | 2360 | 37188 | 40895 | 62010 | 52517 | 112652 | **151877** |
| DoubleDunk | -19 | -16 | 22 | 0 | 22 | -1 | **23** |
| NameThisGame | 2292 | 8049 | 9229 | 13136 | 12761 | 19819 | **28710** |
| Phoenix | 761 | 7243 | 8605 | 108529 | 5327 | 60955 | **367284** |
| QBert | 164 | 13455 | 18503 | 33818 | 14739 | 25712 | **45034** |
| IQM | 0.000 | 1.000 | 1.265 | 3.583 | 1.452 | 4.070 | **7.627** |
| Median | 0.000 | 1.000 | 1.21 | 2.532 | 1.44 | 3.167 | **4.589** |
| Mean | 0.000 | 1.000 | 3.714 | 5.817 | 3.745 | 4.684 | **16.561** |

Table A3: Comparison in terms of performance and walltime against PQN (Gallici et al., 2024). PQN only reports results at 400M frames, and includes life information which has a large effect on performance (see Appendix J). To provide a fairer comparison, we report our results also using life information, but still only use 200M frames. Below are Atari-5 IQM and per-game Scores, with BTR averaged over 3 seeds. For individual games, Human-Normalized scores are reported, with the raw score in brackets.

| Game | BTR (with life info, 200M frames) | PQN (with life info, 400M frames) |
|---|---|---|
| Inter-Quartile Mean | **14.02** | 3.86 |
| BattleZone | **13.53 (473,580)** | 1.51 (54,791) |
| DoubleDunk | **-14 (23.0)** | 6.03 (-0.92) |
| NameThisGame | **4.59 (28,710)** | 3.18 (20,603) |
| Phoenix | **89.95 (583,788)** | 38.79 (252,173) |
| QBert | **14.54 (193,428)** | 2.37 (31,716) |
| Walltime (A100) | 22 Hours | **2 Hours** |
| Backend | (PyTorch (non-compiled) + gymnasium async) | (JAX + envpool) |

## B   FULL RESULTS GRAPHS

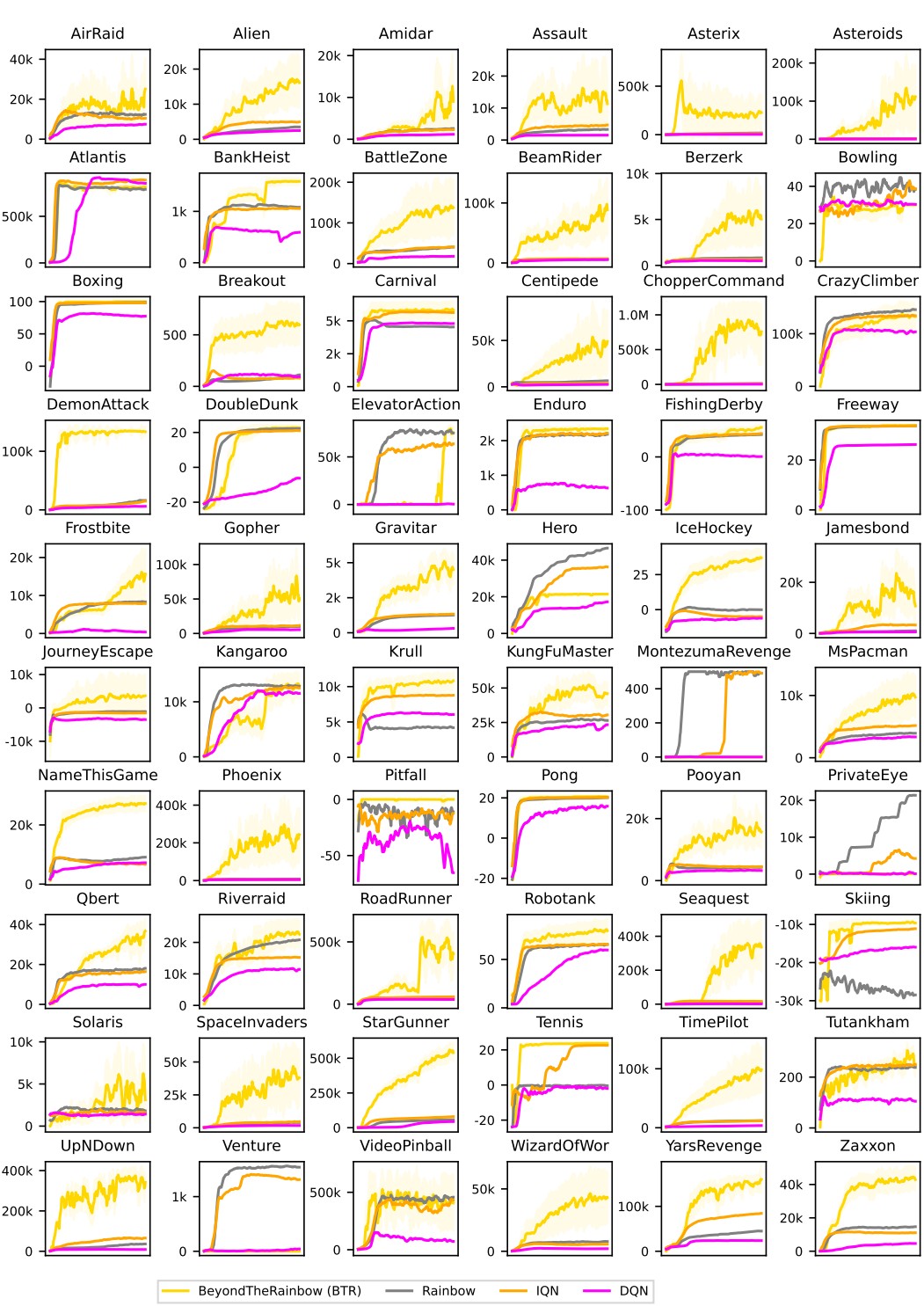

Figure B1: Performance of BTR on each individual game in all 60 Atari games. Results only use a single seed, so may be inaccurate. Shaded areas show 1 standard deviation of scores within that evaluation of 100 episodes.

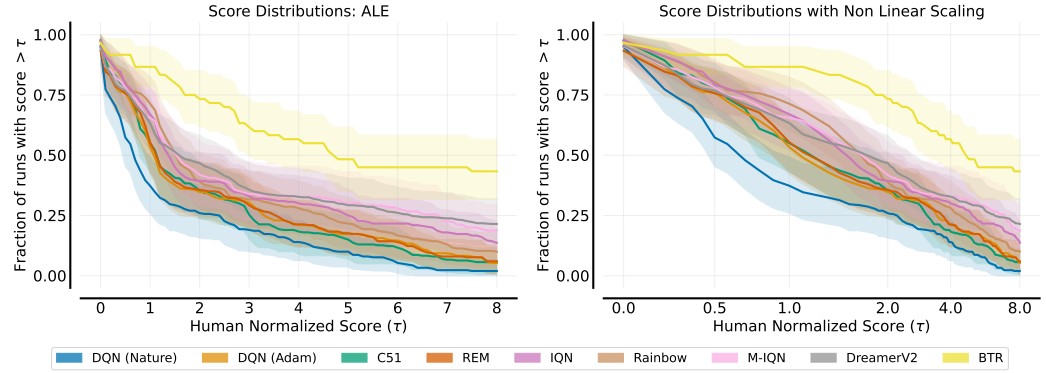

Figure B3: Final performance of BTR on Atari-60 (as used in RLiable (Agarwal et al., 2021)), against other popular algorithms. Plot displays performance profiles, with 95% confidence intervals with task bootstrapping. Please note however that BTR only uses a single seed, and thus these results should be used with care.

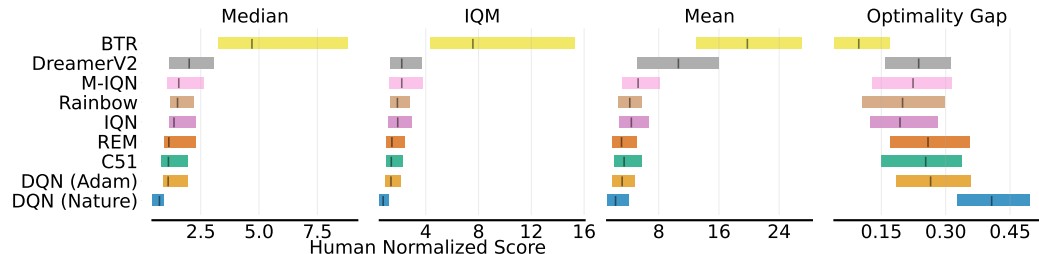

Figure B2: Box plot performance on Atari-60 of BTR against other algorithms reported by RLiable (Agarwal et al., 2021). BTR uses 1 seed, hence large error bars.

## C  HYPERPARAMETERS

### C.1  ENVIRONMENT DETAILS

Table C4: Environment Details for Atari Experiments.

| Hyperparameter | Value |
|---|---|
| Grey-Scaling | True |
| Observation down-sampling | 84x84 |
| Frames Stacked | 4 |
| Reward Clipping | [-1, 1] |
| Terminal on loss of life | False |
| Life Information | False |
| Max frames per episode | 108K |
| Sticky Actions | True |

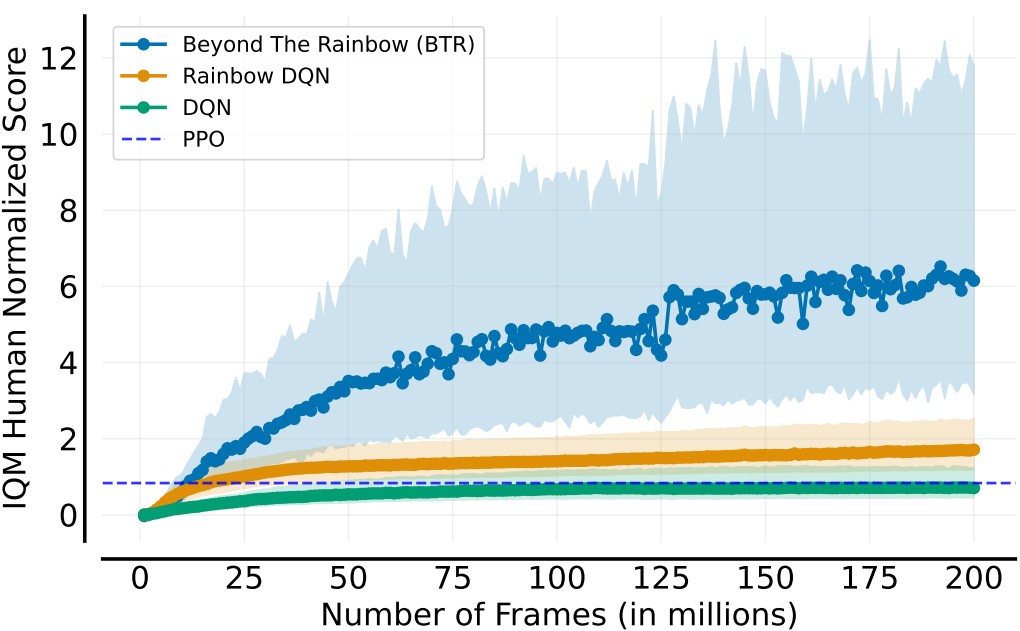

Figure B4: Figure shows BTR against PPO. PPO uses the Cleanba (Huang et al., 2023) implementation, and the plot also only uses the 53 game Atari games Cleanba provides. Shaded areas show 95% confidence intervals with bootstrapping over tasks, however BTR only uses 1 seed, hence the large error size.

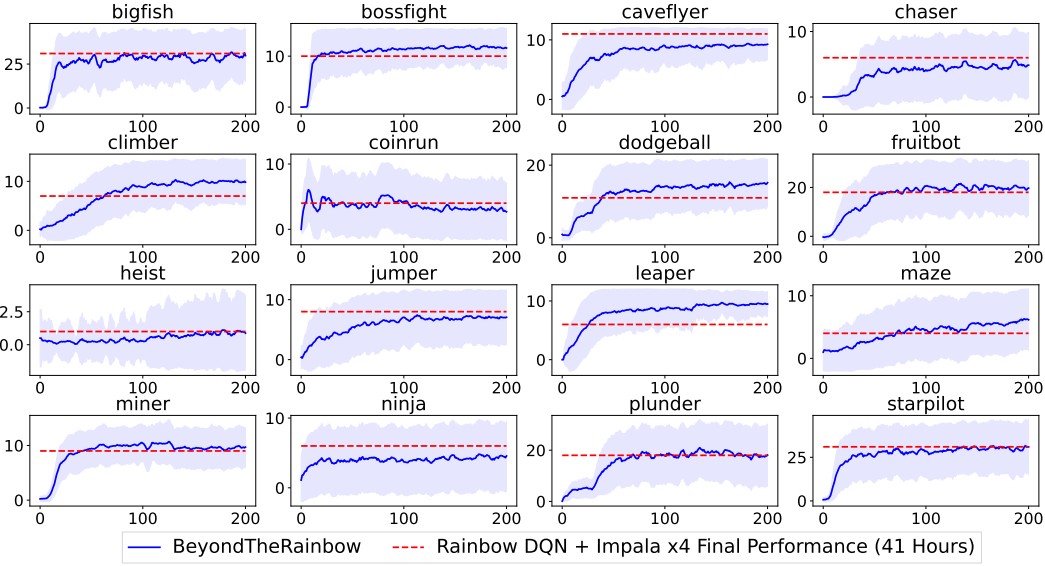

Figure B5: Performance of BTR on each individual game in the Procgen benchmark. Shaded areas show one standard deviation of the performed evaluations. The red dotted line shows the performance of Rainbow DQN + Impala with 4x scaled Impala blocks (Cobbe et al., 2020), after 200M frames.

Table C5: Environment Details for Procgen Experiments.

| Hyperparameter | Value |
| --- | --- |
| Grey-Scaling | True |
| Observation Size | 64x64 |
| Frames Stacked | 4 |
| Reward Clipping | False |
| Max frames per episode | 108K |
| Distribution Mode | Hard |
| Number of Unique Levels (Train & Test) | Unlimited |

## C.2 ALGORITHM HYPERPARAMETERS

## C.3 CLARITY OF THE TERMS FRAMES, STEPS AND TRANSITIONS

Throughout the Arcade Learning Environment's history (ALE) (Bellemare et al., 2013; Machado et al., 2018), there have been many ambiguities around the terms: frames, steps and transitions, which are sometimes used interchangeably. Frames refer to the number of individual frames the agent plays, including those within repeated actions (also called frame skipping). This is notably different from the number of steps the agent takes, which does not include these skipped frames. When using the standard Atari wrapper, training for 200M frames is equivalent to training for 50M steps. Lastly, transitions refer to the standard tuple $(s_t, a_t, r_t, s_{t+1})$, where the timestep $t$ refers to a steps, not frames. We encourage researchers to make this clear when publishing work, including when mentioning values of different hyperparameters.

## D  BEYOND THE RAINBOW ARCHITECTURE & LOSS FUNCTION

### D.1  ARCHITECTURE

Figure D6 shows the the neural network architecture of the BTR algorithm. The architecture is highly similar to the Impala architecture (Espeholt et al., 2018), with notable exceptions:

- **Spectral Normalization**  Within each Impala CNN blocks, each residual layer (containing two Conv 3x3 + ReLu) has spectral normalization applied, as discussed in Section 3.1.

- **Maxpooling**  Following the CNN blocks, a 6x6 adaptive maxpooling layer is added.

- **IQN**  In order to use IQN, it is required to draw Tau samples which are multiplied by the output of the CNN layers, as shown by the section 'IQN Samples' in figure D6.

- **Dueling**  Dueling (as included in the original Rainbow DQN) splits the fully connected layers into value and advantage streams, where the advantage stream output has a mean of 0, and is then added to the value stream.

- **Noisy Networks**  As included in Rainbow DQN, Noisy Networks replace the linear layers with noisy layers.

Lastly, the sizes of many of the layers given in Figure D6 are dependant upon the Impala width scale, of which we use the value 2. For example, the Impala CNN blocks have [16×width, 32×width, 32×width] channels respectively. The output size of the convolutional layers (including the max-pooling layer) is 6×6×32×width, as a 6x6 maxpooling layer is used. Lastly, the cos embedding layer after generating IQN samples requires the same size as the output of the convolutional layers, hence the size is selected accordingly. Another benefit of the 6x6 maxpooling layer is following the product of the convolutional layers and IQN samples, the number of parameters is fixed, regardless of the input size. Figure D7 shows the numbers of parameters the ablated versions of BTR have.

Table C6: Table showing the hyperparameters used in the BTR algorithm.

| Hyperparameter | Value |
| --- | --- |
| Learning Rate | 1e-4 |
| Discount Rate | 0.997 |
| N-Step | 3 |
| IQN Taus | 8 |
| IQN Number Cos' | 64 |
| Huber Loss $\kappa$ | 1.0 |
| Gradient Clipping Max Norm | 10 |
| Parallel Environments | 64 |
| Gradient Step Every | 64 Environment Steps (1 Vectorized Environment Step) |
| Replace Target Network Frequency (C) | 500 Gradient Steps (32K Environment Steps) |
| Batch Size | 256 |
| Total Replay Ratio | $\frac{1}{64}$ |
| Impala Width Scale | 2 |
| Spectral Normalization | All Convolutional Residual Layers |
| Adaptive Maxpooling Size | 6x6 |
| Linear Size (Per Dueling Layer) | 512 |
| Noisy Networks $\sigma$ | 0.5 |
| Activation Function | ReLu |
| $\epsilon$-greedy start | 1.0 |
| $\epsilon$-greedy decay | 2M Frames |
| $\epsilon$-greedy end | 0.01 |
| $\epsilon$-greedy disabled | 100M Frames |
| Replay Buffer Size | 1,048,576 Transitions ($2^{20}$) |
| Minimum Replay Size for Sampling | 200K Transitions |
| PER Alpha | 0.2 |
| Optimizer | Adam |
| Adam Epsilon Parameter | 1.95e-5 (equal to $\frac{0.005}{batchsize}$) |
| Adam $\beta1$ | 0.9 |
| Adam $\beta2$ | 0.999 |
| Munchausen Temperature $\tau$ | 0.03 |
| Munchausen Scaling Term $\alpha$ | 0.9 |
| Munchausen Clipping Value ($l_0$) | -1.0 |
| Evaluation Epsilon | 0.01 until 125M frames, then 0 |
| Evaluation Episodes | 100 |
| Evaluation Every | 1M Environment Frames (250K Environment Steps) |

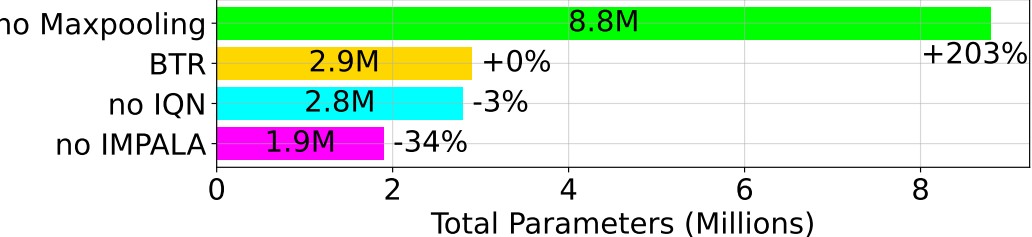

Figure D7: Total number of parameters in BTR with different components removed. Those not included in the graph (Munchausen and Spectral Normalisation) used the same number of parameters as BTR.

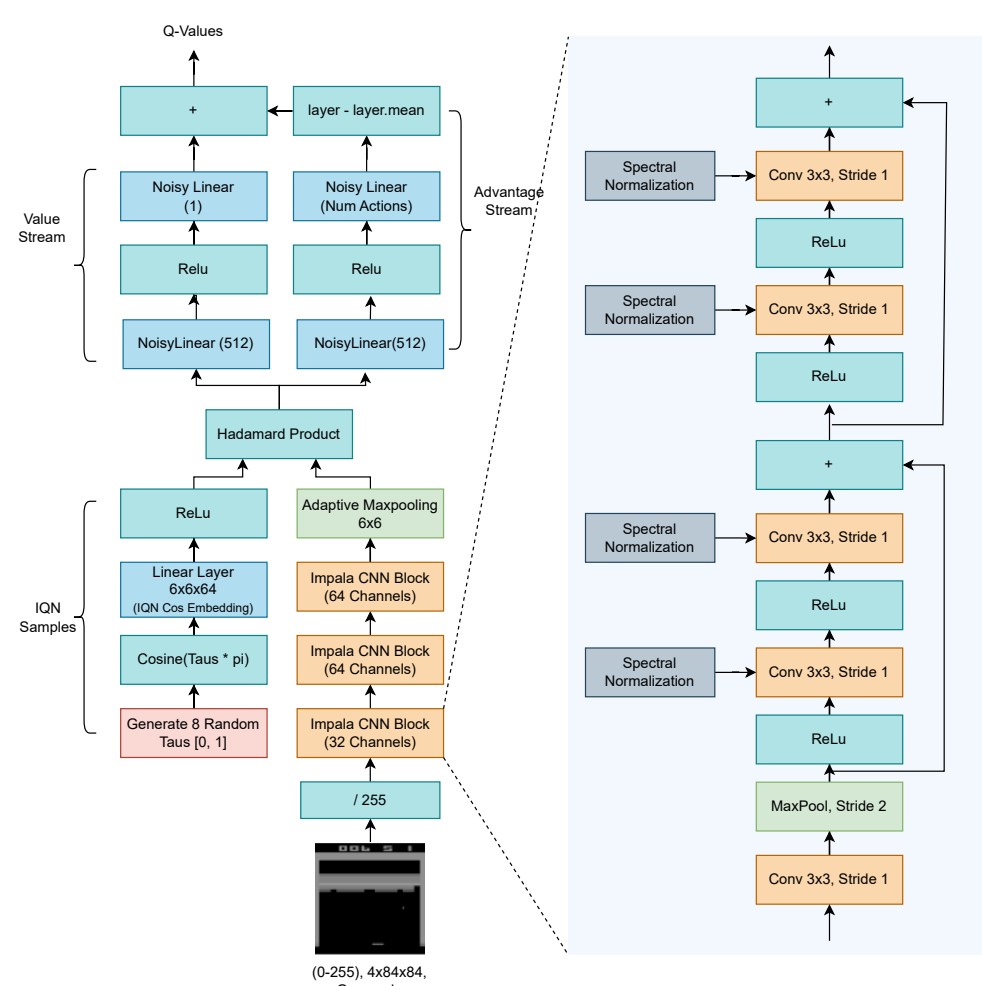

Figure D6: Architectural diagram of the BTR algorithm's neural network. The model contains a total of 2.91 million parameters, 2.52 million of which are within linear layers.

## D.2 LOSS FUNCTION

The resulting loss function for the BTR algorithm remains the same as that defined in the appendix of the Munchausen paper, which gave a loss function for Munchausen-IQN. As the other components in BTR do not affect the loss, the resulting temporal-difference loss function is the same. For self-containment, we include this loss function below:

$$TD_{BTR} = r_t + \alpha[\tau \ln \pi(a_t|s_t)]_{l_0}^0 + \gamma \sum_{a \in A} \pi(a|s_{t+1})(z_{\sigma'}(s_{t+1}, a) - \tau \ln \pi(a|s_{t+1})) - z_\sigma(s_t, a_t)$$

(D1)

with $\pi(\cdot|s) = sm(\frac{\tilde{q}(s,\cdot)}{\tau})$ (that is, the policy is softmax with q̃, the quantity with respect to which the original policy of IQN is greedy). It is also worth noting here that due to the character conflict of both Munchausen and IQN using $\tau$ (Munchausen as a temperature parameter, and IQN for drawing samples), we replace IQN's $\tau$ with $\sigma$. $l_0, \tau$ and $\alpha$ are hyperparameters set by Munchausen. We use the same values in BTR, also shown in our hyperparameter table in Appendix C.2.

# E BTR WITH FEWER TRAINING FRAMES

Some of BTR's improvements provide a relatively small improvement after 200M frames, however we want to point out their importance using fewer samples. Table E7 shows that many improvements provide large benefits earlier in training.

Table E7: A comparison of BTR's ablations when using less than 200M frames on the Atari-5 benchmark. Percentages are given relative to BTR's score.

| Algorithm | 40M Frames | 80M Frames | 120M Frames |
|---|---|---|---|
| BTR w/o Maxpooling | 5.399 (+90%) | 6.895 (+4%) | 7.443 (+2%) |
| BTR | 2.837 ($\pm$ 0%) | 6.613 ($\pm$0%) | 7.297 ($\pm$0%) |
| BTR w/o IQN | 4.030 (+42%) | 3.888 (-41%) | 6.644 (-9%) |
| BTR w/o Spectral Normalisation | 2.145 (-24%) | 5.752 (-13%) | 6.351 (-13%) |
| BTR w/o Munchausen | 2.114 (-25%) | 3.341 (-49%) | 4.755 (-35%) |
| BTR w/o Impala | 0.535 (-81%) | 1.183 (-82%) | 1.372 (-81%) |

# F ANALYSIS OF BTR'S EXTENSIONS

In recent years, several measures have been devised to understand the impact of extensions on a model's behaviour, one of the most popular of which is dormant neurons (Kumar et al., 2020; Sokar et al., 2023). Figure F9 plots their prevalence for each of the ablations. We observe three key changes from the ablations; within the CNN layers, the Nature CNN (as used in Rainbow DQN) has a higher number of dormant/low activation neurons, indicating that the Impala network is significantly better for reducing dormant neurons. As for the fully connected layers, dormant neurons were very high across the board, with IQN lowering the number of dormant neurons.

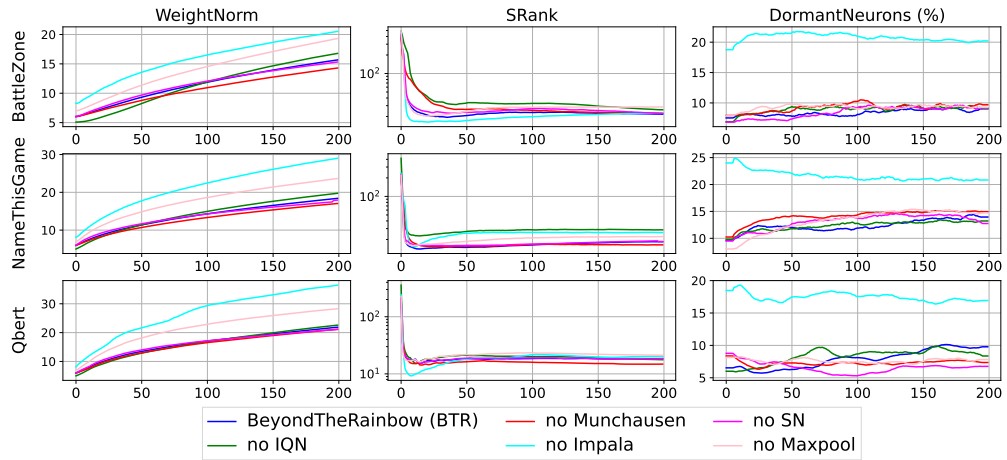

Figure F8: Plot showing L2 norm of network weights, SRank $\delta = 0.01$ (Kumar et al., 2020) and % of dormant neurons (dormant defined as $< 0.1$ (Sokar et al., 2023), for details see Appendix F). Results are based on a single seed so should be used with caution.

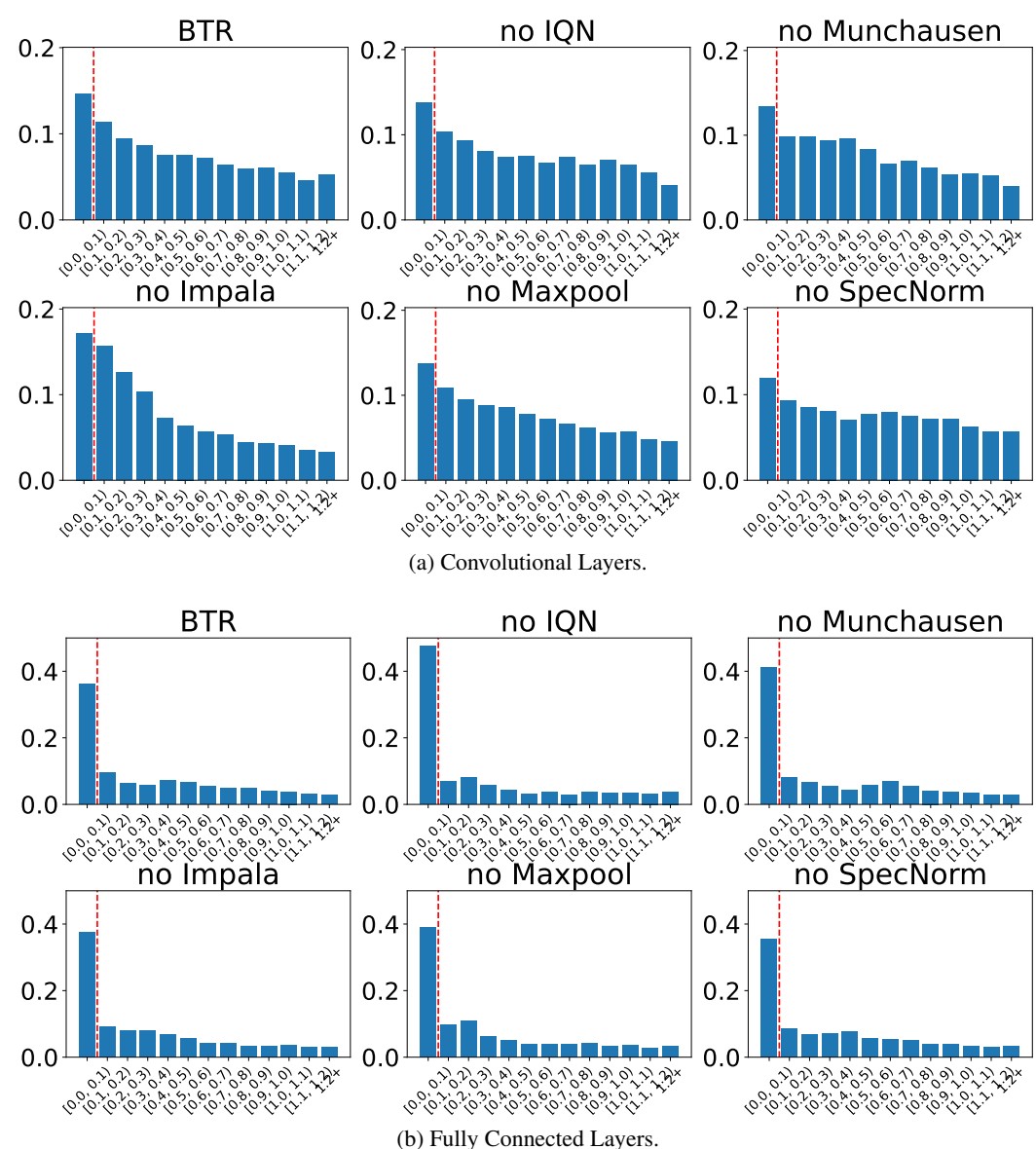

Figure F9: Histogram showing the percentage of average neuron activations for both convolutional and fully connected layers on Atari *BattleZone*, *NameThisGame* and *Qbert*, with subplots for each performed ablation. The first bar in each plot before the red dotted line represent neurons which Sokar et al. (2023) defined as *dormant*. These results are based on a single seed, so should be considered with caution.

## G BTR WITH AND WITHOUT EPSILON GREEDY

One of the first observations we made early in the testing process was that the inclusion of using $\epsilon$-greedy in addition to NoisyNetworks benefited some environments but not others. Specifically, performance was reduced on *BattleZone* and *Phoenix*, both games where the agent reached very high levels of performance with extremely precise control. However, *DoubleDunk* performed significantly worse, only reaching a score of 0, rather than the score of 23 the final BTR algorithm achieved. Similar findings were also found in the full version of Rainbow DQN which used only NoisyNetworks, which achieved a best score of -0.3 (Dopamine's "compact" Rainbow DQN however which did not use NoisyNetworks achieved 22). From this, we conclude that NoisyNetworks

Table F8: Repeat of the main paper's Table 2 for reference against the table of 95% confidence intervals below. Comparison of policy churn, action gaps, actions swaps and evaluation performance with different quantities of $\epsilon$-actions and color jitter (both only applied for evaluation). All measurements use the final agent, trained on 200 million frames, for Atari *Phoenix*, averaged over 3 seeds. Action Gap is the average absolute Q-value difference between the highest two valued actions. % Actions Swap is the percentage of times the agent's argmax action has changed from the last timestep. Policy churn is the percentage of states which the agent's argmax action has changed on after a single gradient step. Color jitter applies a random 10% change to the brightness, saturation and hue of each frame. For associated error with these values, please see Appendix F.

| Category | BTR | w/o Munchausen | w/o IQN | w/o SN | w/o Impala | w/o Maxpool |
|---|---|---|---|---|---|---|
| Action Gap | 0.281 | 0.056 | 0.175 | 0.298 | 0.313 | 0.280 |
| % Action Swaps | 33.4% | 45.8% | 42.2% | 39.7% | 28.6% | 39.2% |
| Policy Churn | 3.8% | 10.2% | 0.5% | 2.9% | 4.2% | 3.9% |
| Score ColorJitter | **206k** | 80k | 93k | 178k | 5k | 172k |
| Score $\epsilon = 0.03$ | **98k** | 47k | 57k | 79k | 5k | 94k |
| Score $\epsilon = 0.01$ | **208k** | 79k | 110k | 181k | 5k | 167k |
| Score $\epsilon = 0$ | 397k | 279k | 199k | 356k | 5k | **489k** |

Table F9: 95% confidence intervals for the main paper Table 2. A repeat of that table is shown above in Table F8.

| Category | BTR | w/o Munchausen | w/o IQN | w/o SN | w/o Impala | w/o Maxpool |
|---|---|---|---|---|---|---|
| Action Gap | [0.28, 0.29] | [0.05, 0.06] | [0.15, 0.2] | [0.27, 0.33] | [0.08, 0.54] | [0.21, 0.35] |
| % Action Swaps | [32.6, 34.2] | [42.2, 49.4] | [38.4, 45.9] | [34.9, 44.5] | [26.0, 31.2] | [38.1, 40.4] |
| Policy Churn | [2.4, 5.2] | [8.3, 12.1] | [0.4, 0.6] | [2.0, 3.9] | [3.3, 5.2] | [2.6, 5.1] |
| Score ColorJitter | [188k, 224k] | [64k, 94k] | [44k, 141k] | [163k, 193k] | [4k, 5k] | [113k, 230k] |
| Score $\epsilon = 0.03$ | [94k, 101k] | [40k, 54k] | [30k, 83k] | [59k, 99k] | [4k, 5k] | [66k, 120k] |
| Score $\epsilon = 0.01$ | [192k, 223k] | [69k, 89k] | [82k, 137k] | [156k, 205k] | [4k, 5k] | [110k, 224k] |
| Score $\epsilon = 0$ | [341k, 452k] | [190k, 367k] | [177k, 220k] | [342k, 369k] | [4k, 5k] | [474k, 503k] |

alone failed to sufficiently explore the environment, whereas $\epsilon$-greedy did not. From these results, we eventually decided to use both methods, but disable $\epsilon$-greedy halfway through training to reap the best of both techniques.

# H  EXPERIMENT COMPUTE RESOURCES

## H.1  OUR COMPUTE RESOURCES

For running our experiments, we used a mixture of desktop computers and internal clusters. The desktop PCs used an GPU Nvidia RTX4090, CPU intel i9-14900k and 64GB of DDR5 6000mhz RAM. When using internal clusters, we used a mixture of GPUs, including Nvidia A100s, Nvidia Volta V100 and Nvidia Quadro RTX 8000. As for CPUs, we used 2 x 2.4 GHz Intel(R) Xeon(R) Gold 6336Y, 48 Cores. Lastly, we saved the models used to produce our analysis, totalling around 300gb across all of our ablations on the Atari-5 benchmark, saving a model every 1 million frames.

As most of our experiments were performed on desktop PC, in the main body of our paper we reference these speeds. We found that desktop PCs actually outperformed internal clusters, likely due to desktop CPUs being more suited to performing environment steps, outlined in the next subsection.

When testing ideas originally (those mentioned in Appendix I), we only tested them using a single run of the games *BattleZone*, *NameThisGame* and *Phoenix* unless otherwise stated. Whilst this method of evaluation is not statistically significant, for preliminary purposes with computational restrictions, we deemed this the best option.

## H.2 BTR WITH DIFFERENT HARDWARE

In this work, we look to make high-performance RL more accessible to those with less compute resources, especially those only with access to desktop computers. Most of our experiments were performed with an RTX4090, we also provide some walltimes for 200M Atari frames for lower-end machines, and provide a brief comparison of desktop PCs against internal clusters:

**Desktops:**

Original: RTX 4090, Intel i9-13900k (2023), 64GB RAM - **11.5 Hours**

RTX 3070, Ryzen 9 3900X (2019), 64GB RAM - **52 Hours**

RTX 2080 ti, Intel(R) Xeon(R) Silver 4112 CPU @ 2.60GHz (2018), 128GB RAM - **32 Hours**

**Internal Clusters:**

Nvidia H100, 48 Core Intel(R) Xeon(R) Platinum 8468 (2023), 2TB RAM - **15 Hours**

Nvidia A100, 24 Core Intel(R) Xeon(R) Gold 6336Y (2021), 512GB RAM - **22 Hours**

We note that there is significant variability in hardware (processors, memory bus speeds, etc), but the results still show reasonable times compared to not using BTR. Overall, we found that training BTR was very capable of running on lower end machines, with the agent (excluding the environments) using around 15GB of RAM. The main performance bottleneck was running the environment in parallel, making the number of CPU cores and processor speed most important. BTR also provides strong performance long before 200M frames, thus providing practical utility for lower-end machines.

## I OTHER THINGS WE TRIED

Throughout the development of the BTR algorithm, we experimented with many different components and hyperparameters. A brief list of ideas we tried that performed worse or equivalent to the final algorithm includes:

Using Exponential Moving Average networks rather than using fixed target networks (this was both computationally slower and performed worse), varying the frequency of updating the target network, changing the size of maxpool layer following the convolutional layers (we tried 4 and 8, however 6 performed significantly better) and decaying the learning rate over the course of training. Using a linearly decaying learning rate from $1 \times 10^{-4}$ to 0 over the course of training gave showed no significant difference. We also experimented with some different learning rates (with and without decay), and found $1 \times 10^{-4}$ to perform best, however $5 \times 10^{-5}$ also performed similarly as was used in Implicit Quantile Networks (IQN). We also tried replacing all ReLu activation units with GeLu, however this lead to dramatically worse performance. Some other ideas which we performed a single-game analysis of included annealing the discount rate from 0.97 to 0.997 (no significant difference on performance), applying spectral normaliation to the linear layers (dramatically worse performance), increasing the number of cos' from IQN (no significant difference on performance) and using Dopamine's Prioritized Experience Replay buffer which doesn't include a $\alpha$ value (moderately worse performance). As discussed in G, we also tried not using $\epsilon$-greedy when using noisy nets.

Recently in RL, it has been shown that Neural Networks have a severe problem with under-parameterisation (Kumar et al., 2020) and neurons becoming dormant (Sokar et al., 2023), preventing larger models from seeing the success evident in other areas of Deep Learning. One method used to remedy this is weight decay through the use of the AdamW optimizer (Loshchilov & Hutter, 2017). As this is far simpler than many other the other techniques to prevent under-parameterisation we decided to test this approach, however no significant differences in performance were observed. We tested this approach using the decay parameter $1e-4$, however potentially using a higher value may results in significant changes in performance.

Lastly we also tried removing some of the original components from Rainbow DQN on Atari *BattleZone*, including Dueling, Prioritized Experience Replay and Noisy Networks. Prioritized Experience Replay and Noisy Networks both proved beneficial, so were kept in the algorithm. Dueling did

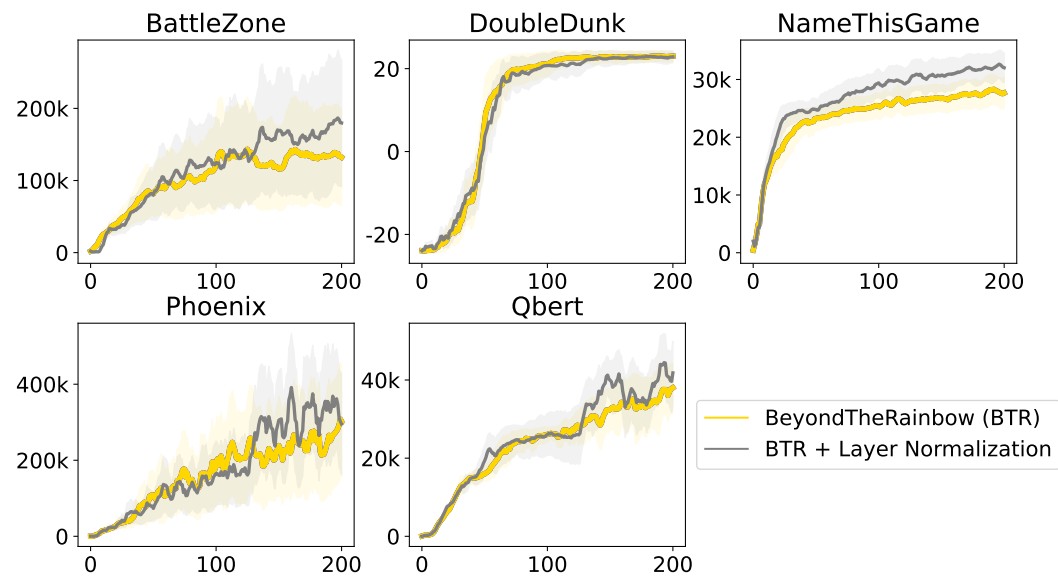

Figure I10: Graph shows individual game performance of BTR with and without Layer Normalization. Layer Normalization makes a notable improvement on multiple games, including *NameThisGame* and *BattleZone*.

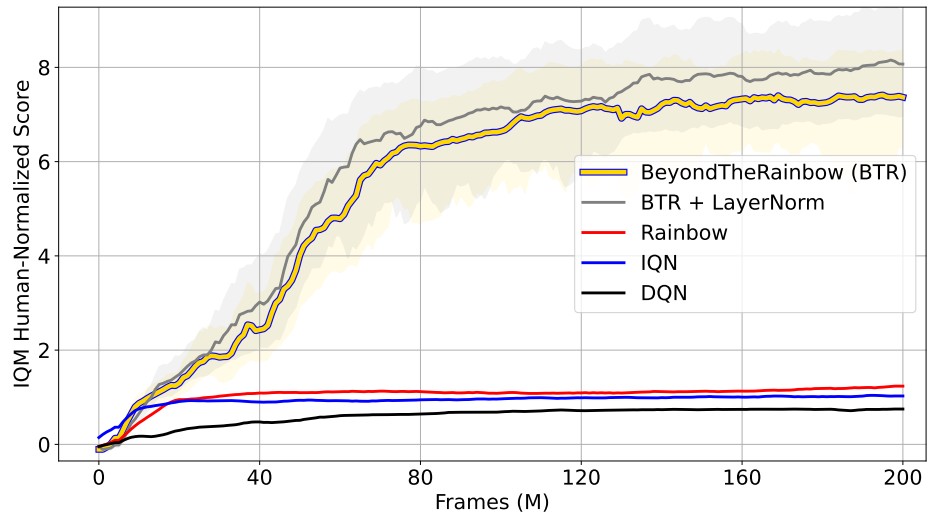

Figure I11: Graph shows IQM human-normalized performance of BTR with and without Layer Normalization on the Atari-5 Benchmark.

not seem to make any significant difference, however we did not choose to remove it for a clearer continuation of Rainbow DQN, in addition to potentially being useful in other Atari environments.

Shortly after the submission of this work, we tested BTR with addition of Layer Normalization, and found positive results. Layer Normalization can improve the robustness to a variety of pathologies that cause loss of plasticity (Lyle et al., 2024), and helps to improve the conditioning of the network's gradients in RL (Ball et al., 2023). Below in Figures I10, I11 and Table I10, we show the results of this addition into BTR.

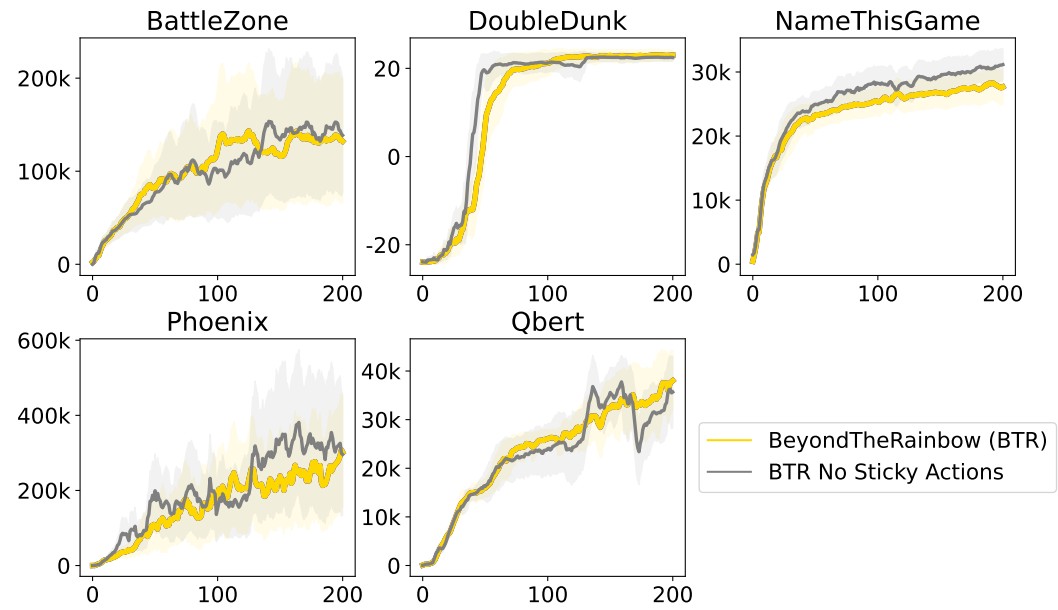

Figure J12: Graph shows individual game performance of BTR with and without Sticky Actions. *NameThisGame* and *Phoenix* see small improvements. When using sticky actions, the impact of disabling $\epsilon$-actions is far more noticeable ($\epsilon$-actions are disabled after 125M frames). This indicates using sticky actions produces a policy more robust to noise, as would be expected.

Table I10: Maximum scores obtained during training (averaged over 100 episodes and all performed random seeds) after 200M Frames on the Atari-5 Environment, compared to BTR with Layer Normalization.

| Game | Random | Human | BTR + Layer Normalization | BTR |
|---|---|---|---|---|
| BattleZone | 2360 | 37188 | **204380** | 151877 |
| DoubleDunk | -19 | -16 | **23** | **23** |
| NameThisGame | 2292 | 8049 | **32834** | 28710 |
| Phoenix | 761 | 7243 | **498264** | 367284 |
| QBert | 164 | 13455 | **48485** | 45034 |
| IQM | 0.000 | 1.000 | **8.369** | 7.627 |
| Median | 0.000 | 1.000 | **5.801** | 4.589 |
| Mean | 0.000 | 1.000 | **21.099** | 16.561 |

## J ALTERED ATARI ENVIRONMENT SETTINGS

In order to investigate the impact of the environmental sticky actions parameter and to compare against other works, we include results for it on the Atari-5 benchmark in Figure J12.

Some prior works choose to pass life information to the agent (Schmidt & Schmied, 2021). To clarify, this is different to terminal on loss of life. Life information does not reset the episode upon losing a life, but does pass a terminal to the buffer, allowing the agent to experience further into episodes while also giving the agent a negative signal for losing a life. This setting is not recommended in Machado et al. (2018), and works which use it are **not** comparable to those which don't. To emphasize this point, we take the three games from the Atari-5 benchmark which use lives (as not all Atari games do), and perform a comparison.

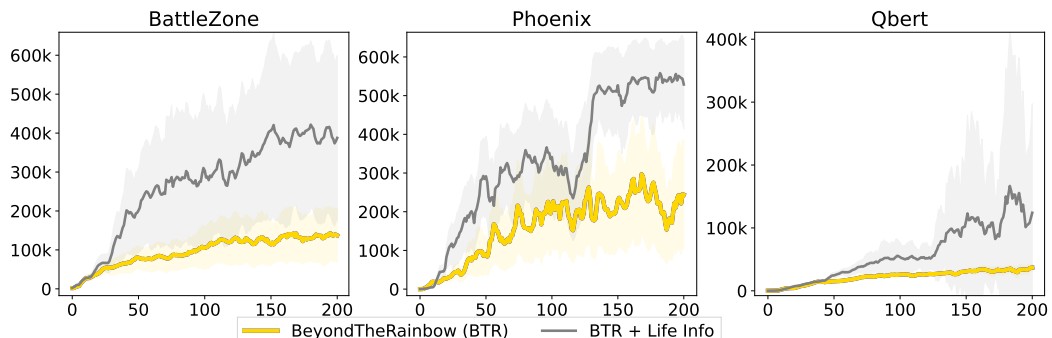

Figure J13: BTR with and without life information.

## K  BTR FOR WII GAMES

BTR interfaces with different Wii Games via the Dolphin Emulator. Specifically, we use a forked repository of to allow Python scripts to interact with the emulator. This includes loading savestates (used to reset episodes), grabbing the screen as a PIL (Clark, 2015) image at the Wii's internal resolution of 480p (downsampled to 140x114 and grey-scaled, used for all observations), reading the Wii's RAM (used for reward functions and termination conditions) and allows programmatic input into the emulator (used for actions). Using Dolphin's portable setting, we are able to run multiple Dolphin Emulators simultaneously on the same machine. Each instance runs as a unique process, and communicates with the agent via Python's multiprocessing library. Similarly to the Atari benchmark, for all games we used a frameskip of 4.

### K.1  SUPER MARIO GALAXY

This environment used Super Mario Galaxy's final level, *The Center of the Universe*, and had to make it to the final fight at the end of the game. The agent had 6 actions, including None, moving in each direction and jumping. Additionally, if the jump action was performed following a movement, the agent would continue to move in that direction.

Rewards were given via finding many values in the Wii's memory that resembled progress in the level. The agent was then rewarded for this progress value increasing from the last frame. If the agent's position entered a set region, the progress variable would be moved. Additionally, the game uses a life system, where the player has a maximum of 3 lives and can lose or gain lives in many different ways. The agent was given a reward of +1 for gaining a life, and -1 for losing a life. Lastly, episode termination occurred if the agent reached 0 lives, or if the agent made it to the end of the level. For this task, we also allowed the agent to start episodes at many points throughout the level, which rapidly sped up training since the agent could easily experience different areas of the level.

Whilst a difficult task, once the agent first completed the level, it did not take long to start consistently completing it due to the deterministic nature of the game.

### K.2  MARIO KART WII

The Mario Kart Wii environment had the agent play against the game's internal opponents (on hard mode, with 12 racers including the agent), on the course Rainbow Road (with items on the 150cc speed setting). The agent had to complete 4 laps of the course to finish the race. The agent had just 4 action, including accelerate, drifting left or right, and using its item. While this limited the agent's potential actions substantially, we found using fewer actions to dramatically accelerate training.

Rewards of +1 were given via reaching checkpoints that were scattered throughout the course (100 in total per lap). Additionally, if the agent's speed dropped below a set threshold (65 km/h), the agent would receive a reward of -0.01 per frame. The agent would be terminated with a reward of -10 if its speed dropped below the threshold for over 80 frames, or with a reward of +10 for finishing the race, with a bonus based on the position the agent finished in. Lastly, the agent was rewarded with a +1 for using its item. Without this reward, we found the agent to often neglect using its item,

likely due to many of the items only providing rewards in the long term, such as slowing down other racers or blocking incoming items far in the future. Similarly to Super Mario Galaxy, we had the agent start the episode in multiple positions around the first lap, allowing it to experience the whole track early in training.

This agent took the longest to train, taking around 160M frames to reach consistent completion. In particular, the agent took a long time to consistently complete the race due to the other racers and randomized items making the environment highly stochastic, with many rare scenarios which could cause the episode to terminate.

### K.3 MORTAL KOMBAT

The Mortal Kombat environment put the agent in the game's *endurance* mode, where the agent would sequentially fight 15 different opponents, but keep retain its health between fights, and only gain health after defeating every 3 opponents. We provided the agent with 14 actions, including: None, Left, Right, Up, Down, Axe Kick, Punch, Snap Kick, Grab, Block, Toggle Weapon, Jump Left, Jump Right, and Crouch. These actions were far from the game's total action space, and limited the agent's ability to perform some of the combos within the game. We limited the agent's actions as the full action space is extremely large.

The agent was positively rewarded for damaging the opponent, and negatively rewarded for taking damage, with one taking one tenth of the health bar equating to +1 reward respectively. The episode was terminated with a reward of -10 for reaching 0 health, and +10 for defeating the 15th and final enemy.

The Mortal Kombat agent learned considerably faster than Super Mario Galaxy and Mario Kart Wii, first completing the environment in 50M frames, and getting progressively more consistent until training was stopped at 90M frames. The agent quickly learned how to dodge enemy hits, and relied heavily upon this strategy.

## L ATARI-5 REGRESSION PROCEDURE

In our main paper ablation figure (Figure 4), we considered using the regression procedure recommended in Atari-5 (Aitchison et al., 2023). This precedure is typically used to predict the Median score across the entire 60 game Atari suite, while only needing to use 5 games. We find that BTR was likely far outside of the distribution that this regression procedure was trained on, given we get very poor predictions when comparing the Atari-5 prediction to the results from running BTR on all 60 games (134.72% relative error at 200M frames). Instead, we opted to just use the IQM across the 5 games to give an easy to interpret average. Figure L14 shows the 60 game suite's true median, compared to the median predicted by Atari-5. These appears to be largely due to Phoenix causing overestimation, where BTR achieves a human-normalized score of 56.55 compared to, for example, Rainbow (Dopamine) with 1.21.

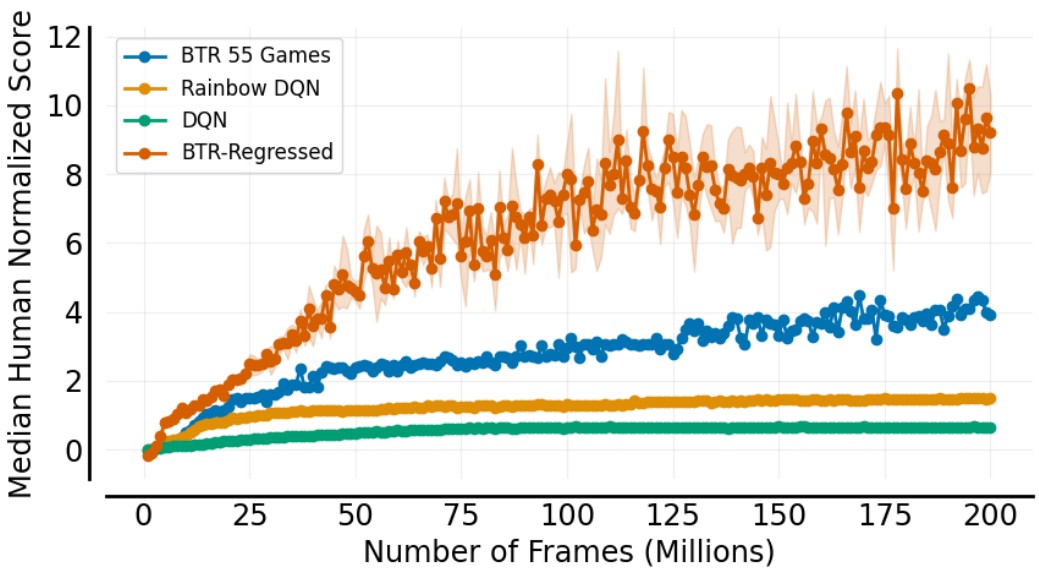

Figure L14: BTR's 60 game median (based on a single seed) against that predicted by Atari-5 (3 seeds). Shaded areas show 95% confidence intervals.

