# OpenReview forum: "Beyond The Rainbow: High Performance Deep Reinforcement Learning On A Desktop PC"
_ICLR.cc/2025/Conference — Submitted to ICLR 2025_

### Official Review · Reviewer_2n46 · 2024-11-01

**Soundness:** 1
**Presentation:** 2
**Contribution:** 3
**Rating:** 3
**Confidence:** 4

**Summary:**

The authors present Beyond the Rainbow (BTR), an algorithm combining advances in Q-learning based approaches to Atari developed since Rainbow.

The authors train their agent on Atari, Procgen and 3 games which aren't well-established benchmarks in RL (Super Mario Galaxy, Mario Kart and Mortal Kombat). They run ablations on their method and demonstrate that the Impala architecture contributes the most to their method's performance. They also demonstrate that vectorization of the environment is key to the faster runtime of their algorithm.

**Strengths:**

The paper has a number of positive points:
- The core idea of trying to achieve strong performance using Q-learning on a desktop PC has significant merit and would constitute an interesting contribution
- The introduction of new games to evaluate on is interesting and the games chosen would make good potential benchmarks.
- The paper is easy to follow and clearly written

**Weaknesses:**

I slightly feel for the authors of this paper. I would like to be able to commend the paper on its empirical results, or the performance on the new environments, but the results are not presented scientifically enough for me to do that, and so I can't recommend acceptance at this venue.

To make my objections more concrete:
- In Figure 2, the authors claim that their method outperforms the red baseline, but this is plainly not the case from the plot. The error bars so significantly overlap the red line there is no way this result is significant.
- The authors do not aggregate their results in accordance with recommended practice [1]. Although they use the inter-quartile mean, they do not provide bootstrapped confidence intervals to estimate errors and do not seem to provide errors in their baseline results. This issue appears in Figures 1 and 2. As far as I know, the authors do not state what the error bars in Figures 1 and 2. If the plotted error bars are standard
- While the evaluation of their method on new games is nice, I can't take any information away from this without even a semblance of a baseline. Training an RL policy on Wii games has no intrinsic scientific value -- it is only by contextualisation of a baseline that this would be a compelling result. Similarly, the authors provide no error bars in this domain.
- Figure 4 again because of the way the results were processed provides almost no information. Atari-5 [2] provides a way to estimate the median given performance on those 5 games. But this is only after the application of a regression procedure. Without the application of this summary metric, it is just not clear what to take away from these results. This figure does not even present human normalised scores, as is standard. This Figure should therefore be replaced by a plot of the regressed median for Atari-5 with bootstrapped confidence intervals. The authors can use rliable [1] for this.
- Again, the analysis in Section 5.2 *should* be compelling and interesting reading, but it's just not done thoroughly enough. Figure 6 is presented without error bars and so are the results in Table 2 and the IQM in Table 3. It's just not possible to believe the authors' conclusions on their work without any estimates of error.
- Additionally, the authors use dormancy [3], but set a threshold of 0.1. Although resetting dormant neurons was shown to improve performance, neurons with a small activation are not in themselves a problem! A neuron followed by a ReLU activation that always outputs 0 is not learning, which clearly constitutes a problem, but a neuron that outputs a small value is still perfectly plastic. The dormancy results therefore also aren't a proxy for any form of plasticity.
- The authors make multiple claims about their method being "state-of-the-art for a desktop PC" (or similar). These should be removed from the paper as they are just impossible to verify. Even as an expert, I do not know the hardware that every paper ran experiments on and whether it would be possible to run it on a desktop PC, and it is not a claim that can be clearly backed-up. I note that the authors did not do all their experimentation on a desktop PC, but only claim that their method can run on one effectively.

**Questions:**

See weaknesses.


Overall, this work is just not good enough in its current format. I recommend that the authors fix the presentation of the results, especially adding error bars and effective aggregation using a tool like rliable. Given the significant problems with every figure and table in the main body of the paper, this work is not good enough for this venue in its current form and would require wholesale changes to fix that.

[1] Deep Reinforcement Learning at the Edge of the Statistical Precipice. Agarwal et al. Neurips 2021.

[2] Aitchison, Matthew, Penny Sweetser, and Marcus Hutter. "Atari-5: Distilling the arcade learning environment down to five games." International Conference on Machine Learning. PMLR, 2023.

[3] Sokar, Ghada, et al. "The dormant neuron phenomenon in deep reinforcement learning." International Conference on Machine Learning. PMLR, 2023.

---

> ### Author Response · Authors · 2024-11-15
>
> We thank you for understanding our core motivation for this work and appreciating what it achieved.
>
> **Error Bars**
>
> In this work, we computed the error bars using the evaluation episode variance rather than as proposed in [1] with the seed variance (for the mean evaluation performance). Using [1], for the Atari-5, we trained BTR across 3 seeds, finding IQM scores of 7.77, 8.22, and 7.86. However, due to time and computing limitations, we only trained BTR for 1 seed on Atari-60 and ProcGen, meaning we can’t use [1] for error bars. If the reviewer agrees, we will train BTR for Atari-60 and ProcGen with three seeds, allowing us to update the error bars. However, due to the computational and time requirements of these experiments, we could not provide these results within the discussion period though they would be ready before the camera-ready version deadline.
>
> Regarding Figure 2, we did not intend to claim that we achieved state-of-the-art performance compared to prior work. Instead, we claim that we achieved similar performance as the prior state-of-the-art with significantly fewer resources, in particular, walltime and learnable parameters. We will change our wording on Line 083 from “outperformed” to “similar” and Line 274 from “exceed” to “similar”. Does this address the reviewer's concern?
>
> **Comparison of prior work on Wii Games**
>
> When demonstrating BTR in these new environments, we aimed to showcase BTR’s capability in games that are not normally considered in RL research. Therefore, we didn’t compare to the prior works shown in Figure 1 that achieved significantly lower scores than BTR. If the reviewer believes this is a crucial comparison to include, we are willing to train a Rainbow DQN agent for the Wii games to include in Figure 3.
>
> **Ablation Study**
>
> We are happy to use human-normalized performance to measure each ablation’s performance and will update Figure 4. Regarding the regression procedure, we are not making claims about the performance of all Atari games from these ablations, rather, we used Atari-5 as a benchmark to better discriminate each ablation’s performance without needing to train on all Atari 60 environments to find similar results. Does this answer the reviewer’s concern?
>
> **Dormant Neurons**
>
> We really appreciate this comment as we ourselves are skeptical of dormant neurons as a measurement. However, we included it as it has become a popular measure in the literature [1,2,3]. Additionally, in Appendix F, we plot a histogram of activations showcasing the range of activations used by the agent presenting a better overview of the agent’s activations. If you feel these results should not be included, we will remove them at your request.
>
> **Claims of the state-of-the-art on a desktop PC**
>
> We recognise that this is a difficult claim to make; however, after surveying the literature, we believe, to the best of our knowledge, that it is true. Part of the reason we make this claim is that few algorithms achieve greater performance than BTR, and those that do, in their reported training requirements, make it clear that it would not be possible to train their agent on a desktop PC. In Table 3, we survey three top-performing algorithms, listing their claimed resources used and identifying several reasons for preventing them from being a fair comparison to a desktop PC: the required resources with numerous CPUs / GPUs, excessive training time over 5 days or more recently GPU VRAM to train a model (commercial GPU have significantly less VRAM than research-grade GPUs). Is the reviewer aware of any training algorithm that could reasonably compete with our claim?
>
> [1] Sokar, Ghada, et al. "The dormant neuron phenomenon in deep reinforcement learning." International Conference on Machine Learning. PMLR, 2023.
>
> [2] Xu, Guowei, et al. "Drm: Mastering visual reinforcement learning through dormant ratio minimization." arXiv preprint arXiv:2310.19668 (2023).
>
> [3] Qin, Haoyuan, et al. "The Dormant Neuron Phenomenon in Multi-Agent Reinforcement Learning Value Factorization." The Thirty-eighth Annual Conference on Neural Information Processing Systems.

---

> > ### Comment · Reviewer_2n46 · 2024-11-21
> > **Thank you for the response**
> >
> > I want to thank the authors for taking the time to respond to my questions and concerns.
> >
> > I want to start by clarifying my point about Atari-5. The point of that paper is that if you evaluate on their 5 environments, **and then perform their regression procedure** you can evaluate a method's median on Atari-60 with reasonable accuracy. It is not that the IQM, or any other metric evaluated on that subset meaningfully tracks performance on the whole set or is a good comparison. The fact that the authors are aware of that work, but do not process their results in accordance with its recommendations is strange to me. In addition to this, you can use rliable to estimate the error **in that regression** by computing bootstrapped confidence intervals. This is a much more effective way of performing this ablation and it's strange the authors didn't do that in the first place.
> >
> > I think that results that only use 1 seed are also incredibly difficult to trust and hence don't think the current procedure is good enough. This means that they don't have a meaningful comparison to another method anywhere in their paper. To make my position extremely clear, **I am not asking that the authors run multiple seeds on atari-60 necessarily**, I know that running such experiments takes a lot of time without significant compute. I would also accept, for example, evaluating their method using Atari-10 and the **correct regression procedure**.
> >
> > The wii games experiments really are not meaningful without a baseline in my opinion. Given the widespread success of RL at solving game environments, its unclear how impressive these results are without a relevant baseline.
> >
> > I think that adjusting the dormant neurons to measure the fraction that *actually* have zero activation across the whole batch would be a meaningful experiment, but I think that the 0.1 threshold is problematic.
> >
> > On the topic of the state-of-the-art claim, I concede that *if the issues with all the experiments were fixed and the values stayed the same*, it would be possible to make some form of SOTA claim like that.
> >
> >
> > I am not willing to raise my score given the serious issues with the results in this paper. As a brief summary:
> > - Figure 1 only uses 1 seed
> > - Figure 2 also only uses 1 seed
> > - Figure 3 has no baseline
> > - Figure 4 reports unnormalised and unaggregated Atari results.
> > - Table 3 has no error bars
> > - Figure 6 also has no error bars
> >
> > In short, there isn't a single figure in this paper that uses error bars and aggregates performance correctly. The authors in their response have quoted the IQM on Atari-5, which also isn't an acceptable way of measuring performance.
> >
> > Every figure in this paper would have to change for me to accept it at this conference. It's just not suitable in its current form. I'm also surprised that the other reviewers have not raised similar issues to those in my review, given the extensive evaluation problems in this paper.

---

> > > ### Author Response · Authors · 2024-11-22
> > >
> > > Thank you for the concise feedback, however, we are alarmed that the reviewer is unwilling to change their score even if we address the comments about the figures. Below, we summarize how we will update each figure to address the reviewers for an updated version of the paper that we aim to submit Monday.
> > >
> > > On the regression procedure of Atari-5, we understood this; however, we were using this subset not to make claims about the possible Atari-55 median performance but rather as a small subset for testing new extensions/improvements. Using our data from training BTR on Atari-55 to the predicted values using the Atari-5 regression procedure, we find that it wildly overpredicts with a median score of 9.20 [8.11,10.23] (95% confidence intervals) compared to the true value of 3.92 for one seed; a relative error of 134.72%.
> > > We identify two problems with using the regression procedure:
> > > 1. It predicts the Median and not IQM (which is standard) scores.
> > > 2. BTR is significantly outside the regression data’s distribution, in particular with Phoenix where BTR achieves a HNS score 46.7 times better than Rainbow (Dopamine) (56.55 compared to 1.21, respectively).
> > >
> > > This huge gap in predicted vs true performance makes us skeptical of using the regression procedure to make strong claims about BTR’s performance.
> > > Could the reviewer clarify if they still believe we should discuss/include the regression procedure within our paper despite the 134.72% error to our known Atari-55 results? We believe that an actual training result across 1 seed (although we are currently running up to 3 seeds) is better than a predicted result which is wildly inaccurate for BTR.
> > >
> > > - **Figure 1**: We are training BTR for three seeds currently for Atari-60 as we don’t believe that the regression procedure will provide accurate results as discussed above.
> > > - **Figure 2**: Has now been updated to use two seeds and RLiable, however, for the final version, we will include three seeds. From these two seeds, taking their best runs throughout training (as is commonly done in full results tables) they achieve an IQM of 0.73 and 0.79 respectively.
> > > - **Figure 3**: For our Mario Kart Environment, we now provide Rainbow DQN as a baseline. For 120 million frames, we find Rainbow averages 3.8 compared to BTR’s 17.5 at the same point in training (single seed). Following the discussion period, we can also provide this baseline for the other two environments.
> > > - **Figure 4**: Has been updated to show human-normalized scores on all individual games. We include a sixth plot of IQM across the 5 games, not using the regression procedure. To reiterate, we are not claiming this to predict the scores over the whole suite, but rather to provide an easy-to-interpret average across the five-game suite, which is not skewed by outliers such as Phoenix.
> > > - **Table 2**: (The reviewer notes Table 3 but we believe this is a typo) now has confidence intervals over seeds, however, we have to put this in the appendix (with a note in the main paper caption) due to the column width of the new table.
> > > - **Figure 6**: Due to high storage demands (saving every model, for every ablation, every million frames, for every seed), we were only able to store the models for a single seed and thus are unable to update the figure. We have, however, updated the caption of the figure to make this clear.

---

> > > > ### Author Response · Authors · 2024-11-25
> > > >
> > > > Please see our updated PDF and list of changes. We believe we have alleviated your concerns regarding Figures 1, 2, 3, 4, 6 and Table 2. Furthermore, we included box plots and performance profiles using RLiable in appendix B.
> > > >
> > > > Below are some key points we think may interest you:
> > > > - We found that even with a 1 and 2 seeds respectively our results outperform other algorithms to 95% confidence, both in Atari and Procgen.
> > > > - We agree Figure 6 did not provide adequate evidence to claim improved plasticity, and thus removed these claims and the figure. We still believe the Figure holds some value, so moved it to Appendix E with a note that it uses a single seed.
> > > > - We use 3 seeds for Table 2, and provide 95% confidence intervals in the appendix (due to table width restrictions). While some of these figures have fairly wide confidence intervals, the claims we have about munchausen and IQN affecting action gaps and policy churn are statistically significant. Regarding our claim that maxpooling helps when evaluating performance under different altered environment regimes, while some figures have fairly large errors, we still find that across 3 seeds, BTR outperforms BTR no maxpooling, compared to the epsilon=0 setting where no maxpooling significantly outperforms BTR. Given this, we believe our claim is not unreasonable.
> > > >
> > > > We thank you for your comments which have improved this work to be at the highest standard of RL. Given that we have fully addressed these concerns, we sincerely hope you are willing to raise your score to reflect this.

---

### Official Review · Reviewer_GWAQ · 2024-11-02

**Soundness:** 3
**Presentation:** 3
**Contribution:** 2
**Rating:** 6
**Confidence:** 4

**Summary:**

This paper combines several different RL improvements to a single algorithm, with a focus on high performance with reasonable computational requirements. In doing so, they find that their approach achieves a new SoTA (not including recurrent methods), while being able to be run on a desktop machine in under a day.

They analyse the factors that led to this performance in detail through several ablations.

Overall, this paper makes rainbow/dqn-type methods more accessible to non-industry labs

**Strengths:**

- This paper gives RL researchers a way to do pretty well in atari without expending too significant computational resources.
- They perform ablations on their individual changes to identify what helps and what has the most effect on performance/walltime. This is quite useful.

I am not giving this a lower score because I think making RL more accessible is worthwhile, and this paper takes a step towards this, and further analyses many of these independent components to see what their effect is. I am not giving a higher score because I think the paper's significance does not warrant it.

**Weaknesses:**

- To me it is unclear if 12 hours is for all games or just 1.
- I wonder how this fits in with the recent trend of hardware-accelerated RL (see e.g., Stoix/PureJaxRL/Gymnax and all of the hardware-accelerated environments). Does that line of work better achieve the goal of making RL more accessible? In that setting, the environment is often run entirely on the GPU, leading to hundreds or thousands of times speedups.

**Questions:**

- The 12 hour number/other timings, is that the total time it takes to train BTR on a single game or on all 57 games?
- It seems like you made quite a few hyperparameter choices (e.g. how often to update, etc.) Do you use the same values for each domain?
- What is the shaded area in the plots? If it is standard deviation it seems like the proposed BTR algorithm is very inconsistent across seeds. Could you elaborate please/maybe provide results for individual seeds?
- Figure 3 does show that you can apply your approach to other games, which is great. I would really like to see some point of comparison, however, to act as a reference point. For instance, run vanilla PPO or DQN or Rainbow as a baseline.
- Why is fig 4 using raw score as the y-axis, as opposed to e.g. normalised?
- Figure 4 is somewhat hard to follow as there are so many lines and it seems like most of them overlap quite a lot.
- Is it feasible to run rainbow with vectorisation? This is not that crucial, it just seems like something obvious to run given figure 5, where vectorisation is the main speedup factor.
- Table 2: Would be nice to have another method, e.g. rainbow or DQN to act as a reference point.
- One recent work that seems to have a similar purpose is "Simplifying Deep Temporal Difference Learning" (https://arxiv.org/pdf/2407.04811). It seems like they use vectorisation as well to achieve large speedups. More importantly, however, is that they primarily use JAX---which is becoming increasingly common in RL, and is reducing computational load significantly/making RL more accessible to compute-poor labs/institutions. Could you please comment on a few things
	- How does this paper's score compare to yours?
	- How does the walltime compare to yours?
	- What do you see as the benefits/disadvantages of this hardware accelerated paradigm compared to the more classic approach you are taking?
- I know it is not usual in these types of papers but I would really appreciate a PPO comparison, both in terms of walltime and performance.

---

> ### Author Response · Authors · 2024-11-15
>
> Thank you for this detailed review with productive and interesting questions.
>
> **Computation and using hardware accelerated environments**
>
> We considered using hardware-accelerated environments, which provide several advantages to researchers, particularly in parallelisation and training speed. However, it is not clear how to hardware-accelerate many real-world problems, e.g., Wii games tested in this work. As we are interested in expanding and investigating RL in a broad variety of challenging environments, we focused on classical CPU-based environments.
>
> We also want to clarify that 12 hours is the time to train an agent for a single environment.
>
> **Different Domains**
>
> Yes, as stated in Section 4.2 (lines 285 to 287), we use the same algorithm hyperparameters for all domains (Atari, Procgen and Wii games). The only thing we changed is the input resolution (140x114 from 84x84) as Dolphin Emulator’s native resolution is much higher, and the number of parallel environments as we found Dolphin was too memory-expensive for more than 4 on a single 64Gb desktop. To counteract this, we took 16 steps (in all 4 environments) before performing an update, which gives us a similar effect to taking 64 steps in parallel, meaning that no additional tuning is required.
>
> **Large Shaded Areas on Plots**
>
> The shaded areas are the standard deviation across all evaluation episodes and seeds. For example, with 100 evaluation episodes and 3 seeds, the shaded area is the standard deviation over all 300 episodes. This tends to be high when performing evaluations due to the surprisingly stochastic nature of atari games (i.e., sticky-actions, no-op starts and long episode lengths), increasing the score variance in a single episode.
>
> BTR’s IQM on Atari-5 for each seed is 7.77, 8.22, and 7.86
>
> **Rainbow with Vectorization**
>
> Adding vectorization to Rainbow is certainly possible, however, it would require extensive tuning to provide a fair and unbiased comparison as BTR’s parameters were tuned to use vectorization (e.g., learning rate, batch size and how often to replace the target network). We are, however, still happy to provide an untuned comparison if you believe this is important.
>
> **Comparison to “Simplifying Deep Temporal Difference Learning” (PQN)**
>
> We think this is a really interesting comparison, and thank you for raising it. Sadly, PQN does not report their full results for 200M frames and importantly uses the life information parameter that alters the environment to provide more regular termination signals, which has been found to significantly improve training. This makes it an unfair comparison to previous works that do not use it. In Appendix J, we evaluate BTR with and without life information so we can provide a fair comparison. See the table below. Additionally, we note that even without life information, at 200 million, BTR still achieves a higher IQM performance at 7.42 than PQN for 400 million with 3.86 across the Atari-5 environments.
>
> Atari-5 IQM and per-game Scores. For individual games, Human-Normalized scores are reported, with the raw score in brackets.
>
> | Game           | BTR (with life info, 200M frames) | PQN (with life info, 400M frames) |
> |----------------|-----------------------------------|------------------------------------|
> | Inter-Quartile Mean | **14.02** | 3.86|
> | BattleZone | **13.53** (473,580)| 1.51 (54,791)  |
> | DoubleDunk  | **14** (23.0) | 6.03 (-0.92) |
> | NameThisGame  | **4.59** (28,710) | 3.18 (20,603) |
> | Phoenix | **89.95** (583,788) | 38.79 (252,173)|
> | Qbert | **14.54** (193,428) | 2.37 (31,716)|
> | Walltime (A100) | 22 Hours (PyTorch (non-compiled) + gymnasium async) | **2 Hours** (JAX + envpool) |
>
> Beyond performance, the largest difference is in wall time for several reasons. First, we use Gymnasium async rather than EnvPool, which is 1.6x faster [1]. Second, we do not use PyTorch compile or cudagraphs, which have both recently been shown to significantly speed up reinforcement learning code which are the equivalent of `jax.jit` [2]. Incorporating them should reduce BTR’s walltime. Third, using a large replay buffer will mean that off-policy algorithms like BTR will be slower from repeatedly accessing RAM than on-policy algorithms like PQN or PPO that can be stored in GPU VRAM. On-policy and off-policy algorithms tradeoff sample efficiency and walltime, which we believe the performance impact is worthwhile for BTR.
>
>
> It is noteworthy that some of BTR’s improvements could be combined with PQN to improve performance, particularly in relation to the neural network architecture.
> We agree that a comparison against PPO in terms of performance and walltime would be interesting, and we will add one in the appendices of our paper.
>
> [1] Weng, Jiayi, et al. "Envpool: A highly parallel reinforcement learning environment execution engine." Advances in Neural Information Processing Systems 35 (2022): 22409-22421.
>
> [2] https://github.com/pytorch-labs/LeanRL

---

> > ### Comment · Reviewer_GWAQ · 2024-11-16
> >
> > Thank you for your prompt response! I will provide a more in depth reply in a few days, but in the meantime:
> > A few points in my original review were not responded to?
> >
> > If you could provide an untuned rainbow with vectorisation comparison that would be stellar.
> >
> > Likewise a PPO comparison would be much appreciated if you have compute.
> >
> > Could you please also comment on why you provide error bars wrt episodes? To the best of my knowledge this is not standard practice.

---

> > > ### Author Response · Authors · 2024-11-18
> > >
> > > Apologies for missing out on a couple of your original points, we will respond to those here:
> > >
> > > We think it would be beneficial to convert Figure 4 to use human-normalized scores as is standard. Furthermore, we also agree that Figure 4 is a little overcrowded, though we wish to include all the ablation options and a couple of comparisons. If the reviewer believes it will help the figure, we are happy to remove the four comparisons (Rainbow DQN with Impala, Rainbow DQN, IQN and DQN) from the main paper and move this comparison to an Appendix.
> > >
> > > On the error bars, due to our limited compute resources, we were not able to run multiple seeds for BTR on Atari-60 or ProcGen before submission, though we have three seeds for Atari-5. We are currently running these additional experiments and will update the figures using the standard practice however the time required will mean that this will be finished after the discussion period. We will however give an example of what this will look like by providing a figure in our PDF of Atari-5 with the updated error bars.

---

> > > > ### Comment · Reviewer_GWAQ · 2024-11-21
> > > >
> > > > Thank you for your response.
> > > >
> > > > I think you can keep figure 4 as is, it is just somewhat hard to parse (although maybe a simplified version in the main text alongside the full one in the appendix could be an option).
> > > >
> > > > I also appreciate the comparison with PQN, and I think it would be beneficial to add it to the paper (in the appendix would be fine)
> > > >
> > > >
> > > > I think the following are still big problems:
> > > >
> > > > 1. Having only one seed is quite problematic, and the aggregation method is nonstandard. This was not made clear in the paper (at least to my reading).
> > > > 2. No baselines for the Wii games is not great: Having at least something is necessary to compare the proposed algorithm against.
> > > > 3. I think it would still be beneficial to compare against PPO, but a more reasonable number of seeds is a higher priority.

---

> > > > > ### Author Response · Authors · 2024-11-25
> > > > >
> > > > > Please see our updated PDF and list of changes.
> > > > >
> > > > > Regarding your previous comments:
> > > > > - While Atari-60 uses one seed, we are in the process of running another 2 seeds, and we already outperform other algorithms such as Rainbow DQN with over 95% confidence.
> > > > > - We have now provided a baseline for Mario Kart Wii. Before the final version, we are also running this for Mortal Kombat and Super Mario Galaxy.
> > > > > - We have added a Table to Appendix A with a comparison against PQN (the same Table as the previous comment).
> > > > > - We now provide a Figure in Appendix B with a comparison against PPO, using results from Cleanba [1].
> > > > > - For Figure 4, we now include an IQM plot to help interpret the figure.
> > > > >
> > > > > [1] Huang, Shengyi, et al. "Cleanba: A Reproducible and Efficient Distributed Reinforcement Learning Platform." The Twelfth International Conference on Learning Representations. 2023.

---

> > > > > > ### Comment · Reviewer_GWAQ · 2024-11-30
> > > > > >
> > > > > > Thank you for your response and improvements to the paper. I will maintain my score of 6, primarily due to the significance of the work. I still advocate for acceptance.

---

### Official Review · Reviewer_CULH · 2024-11-03

**Soundness:** 2
**Presentation:** 2
**Contribution:** 3
**Rating:** 5
**Confidence:** 4

**Summary:**

The paper presents a variant of Rainbow that adds further architectural and algorithmic improvements to improve not only the agent's score but also to increase its training speed to around 3x what has been previously reported, while running it on top-notch consumer hardware. Finally the authors also show that their improved version of rainbow can deal with modern games with complex graphics and physics.

**Strengths:**

The presentation is overall clear, the methodology is sound, and the results are compelling. Both extensive use of ablations, and the connection to other important metrics related to pathologies in Deep RL algorithms are an example that more papers should follow.
The appendices are also data rich, showing ablations' performances on each of ALE's 60 games, and even having one appendix about things that were tried but did not lead to improvements in performance, which may help others not repeat the experiments.

**Weaknesses:**

1. Adaptive Maxpooling is never defined. It's not a common layer in reinforcement learning and it's never defined in the paper, in fact skimming (Schmidt and Schmied, 2021) that layer is also not defined, I believe this is the only seious weakness in the paper's presentation, but still I believe it is a serious weakness (though hopefully the authors can fix it and so I can increase their grade).
2. There are at least 2 relevant citations missing, "Spectral Normalisation for Deep Reinforcement Learning: An Optimisation Perspective" when talking about Spectral Normalisation, and "On the consistency of hyper-parameter selection in value-based deep reinforcement learning" when talking about the need for tuning Deep RL hyperparameters and the benefits of using layer norm between dense layers.
3. I believe it's slightly misleading to not specify "a high-end PC" when talking about the kind of machine that can run the algorithm in 12 hours (4090 RTXs are quite expensive, and i9s are Intel's high-end consumer line)
4. I believe a more direct comparison with Schmidt and Schmied, 2021 is warranted, given its foundational importance to the paper.
5. Using only 3 seeds while having a large increase in the number of tuned hyperparameters weakens the validity of the results as explained in "Empirical Design in Reinforcement Learning", though at the same time the analysis of metrics beyond simply the score and the extensive use of ablations help.

**Questions:**

1. What exactly is adaptive maxpooling? Would it be possible to add a description of it with either an equation, pseudo-code, or diagram?
2. Where did the formula 0.05/batch_size for Adam's epsilon come from?
3. The final algorithm has a considerable number of hyperparameters, would it be possible to discuss a bit which ones are the most important to tune should someone try to apply this algorithm to a new domain?

**Details Of Ethics Concerns:**

I am just slightly worried about the use of somewhat modern Nintendo games as RL environments through the use of emulators, is the use of emulators for research legal?

---

> ### Author Response · Authors · 2024-11-15
>
> Thank you for your diligent and detailed response, which summarizes the paper’s strengths and raises productive criticisms.
>
> **Details of Adaptive Maxpooling**
>
> Thanks for spotting this. We will update the paper to clarify Adaptive Maxpooling and how it works.
>
> Adaptive Maxpooling is identical to the standard maxpooling layer, except the output shape is an argument with the kernel size and stride automatically adjusting for different input resolutions, enabling support for different resolutions with no algorithmic change or needed learnable parameters. Our code simply uses the PyTorch Adaptive Maxpooling 2D layer with a maxpool size of (6, 6) as a drop-in replacement for standard maxpooling.
>
> **Citations and comparison against other work**
>
> In an earlier draft, we included a comparison against Schmidt and Schmied, 2021 in Figure 4, however, after some investigation found that they used a “life information signal” when evaluating their results. As demonstrated in Appendix J, this significantly impacts performance and is not recommended for evaluation by Machado et al., 2018 [1]. Because of this, we decided not to include these results in the main paper as we believed it would confuse readers as to the reason for the performance difference, however, we did still include a comparison in Appendix A.
>
> With regard to your requested citations, we think these are strong relevant additions to the paper and have added them in Sections 3.1 and 3.2, respectively.
>
> **Adam’s Epilson Formula**
>
> The parameter `1.95e-5` for Adam’s epsilon comes from Schmidt and Schmied, 2021. We will add a reference to this. While the value was correct, we apologize for a minor mistake in our formula, which should have been `0.005 / batch_size`, as opposed to `0.05 / batch_size`.
>
> **BTR’s hyperparameters on different domains**
>
> For new domains, e.g., the Wii games, we did not do any further hyperparameter tuning, keeping the same set across Atari, ProcGen, and Wii games. We wouldn’t be surprised if further improvements could be made to individual environments with further tuning, but we aim to produce a single algorithm that works across many domains, which is in line with DQN’s original motivation. In terms of the most important parameters, we found the learning rate, discount rate, and N (from N-Step TD learning) to be the most significant.
>
> **Using A High-End Desktop PC**
>
> We agree about specifying desktop components and will specify them in the introduction. We would like to note that research servers with an Nvidia A100, a four-year-old GPU model, cost over \\$10,000 each, compared to an equivalent high-end desktop used in this research, which costs less than \\$5,000 in total. Additionally, with future consumer graphics cards, we anticipate the cost of building similar desktops will decrease, making this research more accessible.
>
> **Ethical issue with regards to using Wii Games**
>
> On the copyright-based and emulator question for using Wii-based games, we view this as similar to the use of Atari games in RL research, where the ROMs are similarly under Atari's copyright, though they are not viewed as an ethical issue for research. Likewise, we have bought the ROMs for the respective games used and are not distributing the ROMs with the research, so we do not believe this is an issue. If the reviewer is unsatisfied with this response, could they clarify in what way they believe there is an ethical issue here that does not exist for RL research using the Atari games? Furthermore, we have discussed using the Dolphin Emulator with the developers, who have cleared us for use in research.
>
> [1] Machado, Marlos C., et al. "Revisiting the arcade learning environment: Evaluation protocols and open problems for general agents." Journal of Artificial Intelligence Research 61 (2018): 523-562.

---

> > ### Author Response · Authors · 2024-11-25
> >
> > Please see our updated PDF and list of changes.
> >
> > The points which specifically address your comments are listed below:
> >
> > - We now provide more detail in the description of Maxpooling.
> > - We now specify “high-end desktop” in both the abstract and introduction
> > - We have now updated the Adam epsilon formula to `0.005 / batch_size`.
> >
> > Given that we have addressed your concerns, we hope you raise your score accordingly, as you mention in your original comment.

---

> > > ### Comment · Reviewer_CULH · 2024-11-25
> > >
> > > Thank you for the changes. I had not realized how so many of the results are based on a single seed, this almost makes the results in the paper meaningless I'm afraind.
> > > Would it be possible to update the figures in the appendix to use the median of 3 seeds + a tolerance interval for the shaded area? Also I don't believe you define $\epsilon$-actions in the paper.
> > > I still find the paper to have potential, but things like Figure 1, B1, and F8 are again almost meaningless without the use of multiple seeds

---

> > > > ### Author Response · Authors · 2024-11-26
> > > >
> > > > While Figure 1 does use a single seed, we would like to point out that this is an average of 60 tasks, and even our lower 95% confidence intervals substantially outperform other popular algorithms such as Rainbow DQN (see Figure B2). Furthermore, we are currently running additional seeds, and will have 3 completed seeds before the final submission. We will additionally run 3 seeds for Figure F8, however have removed claims about this Figure as we don’t believe they are key to the paper. Given this, we don’t significantly rely on any results we are yet to produce.
> > > >
> > > > Thank you for pointing out the issue with ϵ-actions, we will add this in the section 2.2.

---

> > > > > ### Comment · Reviewer_CULH · 2024-12-02
> > > > >
> > > > > It's not clear from the picture that your lower 95% confidence interval substantially outperforms Rainbow (from the figure they seem very close in fact), which only makes it clearer that without the 3 seeds results we can't be particularly sure whether it's indeed capable of consistently outperforming Rainbow.
> > > > > I like the results of the paper and many of the experiments, but I still don't believe the experimental methodology is up to the standard expected at Neurips, I would recommend the authors study "Empirical Design in Reinforcement Learning" in depth and apply its methods to make the paper strong before re-submitting it to another conference.
> > > > > Given all that I will keep my score

---

> > > > > > ### Author Response · Authors · 2024-12-02
> > > > > >
> > > > > > We appreciate the reviewer’s insistence on high-quality empirical data and are running two more BTR seeds to validate our initial results. For this second seed, we have run 38 games so far, with the newly computed confidence intervals for these games, as shown in the table below.
> > > > > >
> > > > > > | Algorithm         | IQM   | Lower 95% CI | Upper 95% CI |
> > > > > > |-------------------|-------|--------------|--------------|
> > > > > > | BTR (2 seeds)     | 5.480 | 5.049        | 5.928        |
> > > > > > | Rainbow (5 seeds) | 1.533 | 1.500        | 1.568        |
> > > > > > | DQN (5 seeds)     | 0.871 | 0.842        | 0.901        |
> > > > > >
> > > > > > The reason for this significant change in the confidence interval is how CI is computed over a single vs multiple seeds. For a single seed, task bootstrapping describes an algorithm’s sensitivity to tasks, compared to a larger unknown population of tasks. Computing CIs over seeds, however, simply compares how the performance is likely to change over multiple runs.
> > > > > >
> > > > > > Additionally, this second seed outperforms our first seed with IQMs of 5.908 and 5.534. In conclusion, we have shown that BTR outperforms Rainbow to statistical significance, therefore alleviating the concerns of our reviewers.

---

### Official Review · Reviewer_LxNJ · 2024-11-04

**Soundness:** 3
**Presentation:** 3
**Contribution:** 3
**Rating:** 8
**Confidence:** 3

**Summary:**

This paper introduces Beyond The Rainbow (BTR), a novel reinforcement learning (RL) algorithm that enhances Rainbow DQN by integrating six key improvements. The BTR algorithm is computationally efficient, capable of training powerful agents on a standard desktop computer within a short time. Experimental results show that BTR outperforms state-of-the-art RL algorithms on both the Atari-60 and Procgen benchmarks. Additionally, BTR can handle training agents for challenging levels in complex, modern games. Finally, this paper includes a comprehensive ablation study to analyze the performance and impact of each component within the BTR algorithm.

**Strengths:**

1. This paper is well-written and well-organized. The ideas are clear and could be easily understood
2. The experiments are comprehensive and the results are strong. As shown in Section 4, the proposed BTR algorithm could greatly outperform state-of-the-art baselines in two classic benchmarks and handle three hard and complex modern games with a desktop PC.
3. The paper includes extensive ablation studies and experimental data. Section 5 presents a detailed analysis of the performance and impact of each component of the BTR algorithm, providing readers with insights into the sources of the algorithm's performance gains. Additionally, the authors include complete experimental results and settings in the appendix, helping to clarify any potential confusion or misunderstanding for readers.

**Weaknesses:**

1. The BTR integrates six improvements from existing RL literature to Rainbow DQN. While the algorithm demonstrates strong performance, its novelty might appear limited.  Could you further clarify the novelty of this work? Or specifically, could you briefly discuss if there is any challenges in integrating these existing improvements into the BTR algorithm?

**Questions:**

See weaknesses.

---

> ### Author Response · Authors · 2024-11-15
>
> Thank you for your insightful review, which succinctly outlined our paper’s contributions, results, and analysis.
>
> In answer to where the novelty in this work comes from, we identify three sources:
>   - The process of choosing, integrating, and testing components together is a time-consuming process that requires diligence. In addition to the six improvements included in BTR, several more promising extensions were considered and investigated, though dismissed (see Appendix I for more detail). In publishing the paper, we hope this will prevent future researchers from recomputing the same tests conducted in this paper. While integrating these components, we found it necessary to re-tune hyperparameters, especially when using vectorization with components designed to be used with a single actor (e.g., batch size, learning rate, and frequency of updating the target network).
> - We analyze the components across several measures beyond performance, demonstrating their different and sometimes competing effects on action gaps, policy churn, and observational noise. Such analyses have been completed previously, though not across as many measures or as varied improvements.
> - We demonstrate BTR beyond the standard testing suite to Wii games, showcasing BTR’s continued strength as an algorithm outside of standard research testing environments. This includes games with far more graphically intensive and challenging domains than what any other algorithm (with similar compute resources) has been shown to handle.
>
> On the challenge of integrating these components, we found it technically complex to simultaneously support Impala, Spectral Normalization, Maxpooling, Dueling Networks, Noisy Networks, and IQN, which has never before been attempted. An additional complexity was implementing a replay buffer that supports vectorization, N-Step TD Learning and Prioritization in a memory-efficient way. This approach stores each frame individually rather than all four frames together, reducing memory overhead from 28GB to 7GB in total for a million Atari observations. To help future researchers and hobbyists, we open-source our code.
>
> We are happy to further elaborate on any of these at your request.

---

### Author Response · Authors · 2024-11-15

We thank all the reviewers for their responses and the time and effort spent on the reviews, recognising the value of the paper’s motivations and empirical results. Furthermore, we appreciate the insights and constructive criticism that we will use to continue improving the paper.

Towards the end of the discussion period, we will provide an updated PDF of the paper, along with a summary of the changes made.

---

> ### Author Response · Authors · 2024-11-25
>
> Thank you again to all the reviewers for their time and diligence. In the updated paper we have addressed all the reviewers' concerns, now proving statistically significant performance improvements over even more algorithms. In publishing, we hope this research will be able to increase the wider accessibility of high performance RL.
>
> Following the reviewer’s comments, we have made the following changes, which can be found in our updated PDF.
> - We have updated all relevant figures in the paper to use 95% confidence intervals in accordance with the current research standards from RLiable. For Figure 2 and Table 2, we now use 2 and 3 seeds, respectively.
> - For Figure 3 with the Wii games, we have added a baseline for Mario Kart with Rainbow DQN, achieving a score of 3.4 at 155 million frames compared to BTR’s score of 54 at the same point.
> - We have added a more detailed description of Maxpooling and citations for Spectral Normalization and Hyperparameter tuning, as requested by Reviewer CULH.
> - We have clarified “High-end” PC in both the abstract and introduction, as requested by Reviewer CULH.
> - We have updated Appendix C2 with the correct formula for Adam’s Epsilon.
> - In Appendices A and B, we added comparisons to PQN and PPO as requested by Reviewer GWAQ.
> - Figure 4 now includes a comparison to Rainbow + Vectorization, as requested by Reviewer GWAQ.
> - Appendix B includes the Optimality Gap, Performance Profile and Box Plots for BTR from RLiable to be at the highest standard of RL research.
> - Figure 6 has been moved to the appendix, and some claims about plasticity have been removed due to concerns raised by Reviewer 2n46.
>
> Additionally, we are still running BTR for Atari-60 for an additional two seeds. Due to the confidence interval statistics, we expect this to significantly shrink the confidence intervals in Figure 1 to match closer to the error bars in Figures 2 and 3. Lastly, we will also provide Rainbow DQN comparisons for all Wii games.
>
> We look forward to hearing any more comments regarding the updated PDF.

---

### Meta-Review · Area_Chair_QzYL · 2024-12-16

**Metareview:**

This paper proposes Beyond The Rainbow (BTR) an RL algorithm combining multiple recent advancements into a single method with the single aim of outperforming on the Atari benchmark. There is a fairly high bar for this type of work to be accepted, given that the Atari environments are no longer near the forefront of our field and it has been reported multiple times that experimental results in this domain do not necessarily translate. Indeed, the lack of experimental rigor is ultimately the issue that inclines me to recommend rejection for this paper, in particular the lack of seeds.

**Additional Comments On Reviewer Discussion:**

There was a reasonable discussion, the reviewers on the negative side reiterated their views due to the lack of experimental rigor.

---

### Decision · Program_Chairs · 2025-01-22

Reject